# Antigen receptor control of methionine metabolism in T cells

**Linda V Sinclair[1]\*, Andrew JM Howden[1], Alejandro Brenes[2], Laura Spinelli[1], Jens L Hukelmann[2], Andrew N Macintyre[3†], Xiaojing Liu[3], Sarah Thomson[1], Peter M Taylor[1], Jeffrey C Rathmell[4], Jason W Locasale[3], Angus I Lamond[2], Doreen A Cantrell[1]\***

[1]Cell Signalling and Immunology, University of Dundee, Dundee, United Kingdom; [2]Centre for Gene Regulation and Expression, University of Dundee, Dundee, United Kingdom; [3]Pharmacology and Cancer Biology, Duke University, Durham, United States; [4]Center for Immunobiology, Vanderbilt University Medical Center, Nashville, United States

**Abstract** Immune activated T lymphocytes modulate the activity of key metabolic pathways to support the transcriptional reprograming and reshaping of cell proteomes that permits effector T cell differentiation. The present study uses high resolution mass spectrometry and metabolic labelling to explore how murine T cells control the methionine cycle to produce methyl donors for protein and nucleotide methylations. We show that antigen receptor engagement controls flux through the methionine cycle and RNA and histone methylations. We establish that the main rate limiting step for protein synthesis and the methionine cycle is control of methionine transporter expression. Only T cells that respond to antigen to upregulate and sustain methionine transport are supplied with methyl donors that permit the dynamic nucleotide methylations and epigenetic reprogramming that drives T cell differentiation. These data highlight how the regulation of methionine transport licenses use of methionine for multiple fundamental processes that drive T lymphocyte proliferation and differentiation.
DOI: https://doi.org/10.7554/eLife.44210.001

**\*For correspondence:**
l.v.sinclair@dundee.ac.uk (LVS);
d.a.cantrell@dundee.ac.uk (DAC)

**Present address:** †Human Vaccine Institute, Duke University, Durham, United States

**Competing interests:** The authors declare that no competing interests exist.

## Introduction

T lymphocytes responding to antigen undergo rapid proliferation as they differentiate to produce effector populations. T cells also undergo large-scale, dynamic transcriptional remodelling during differentiation in response to immune activation (*Sen et al., 2016*; *Crompton et al., 2016*; *Gray et al., 2014*; *Zhang et al., 2014*). Immune activation of T lymphocytes requires that these cells adapt their metabolic programs to support rapid clonal expansion and cell differentiation. For example, activated T cells accelerate glucose metabolism to fuel oxidative phosphorylation, glycolysis and the production of UDP-GlcNAc to allow critical intracellular protein O-GlcNAcylations (*Donnelly and Finlay, 2015*; *Swamy et al., 2016*). Activated T cells also dramatically upregulate lipid production programs to meet the demand for membrane biosynthesis associated with growth and proliferation (*Kidani et al., 2013*). T cells also need a supply of glucose, leucine and arginine to sustain the activity of the serine/threonine kinase complex Mammalian Target of Rapamycin Complex 1 (mTORC1) (*Finlay et al., 2012*; *Nicklin et al., 2009*; *Wang et al., 2015*), a critical kinase that regulates the differentiation and migratory capacity of effector T cells (*Delgoffe et al., 2011*; *Sinclair et al., 2008*); serine is required for biosynthesis of purine nucleotides needed for T cell proliferation (*Ma et al., 2017*).

It is thus increasingly recognised that ensuring that T cells have a sufficient supply of metabolic substrates for key biological processes is critical for T cell participation in adaptive immune

**eLife digest** White blood cells known as T cells are an essential part of the immune system. If these cells do not work properly the immune system falls down, leading to disease and eventually death. T cells have receptors on their surface that can detect molecules that do not belong in the body and that may indicate an infection or cancer. When one of these foreign molecules is detected, the T cell will activate and transform to become better equipped to protect the body. These two processes known as activation and differentiation involve extensive changes within the T cell. Many of these changes rely on the addition of a small chemical tag onto molecules such as DNA, RNA or proteins. The tag, a methyl group, is most often obtained by breaking down a chemical called methionine, one of the building blocks of proteins.

Like all other animals, humans cannot make methionine, and so we must instead obtain it through our diet or recycle it from existing proteins. This raised some questions: does the availability of methionine limit the activity of T cells? And, if so, is it the uptake or breakdown of methionine that has the biggest effect?

Sinclair et al. answered these questions by studying T cells from mice. First, the T cells were activated, the proteins from those cells were then collected and quantified using a technique called high resolution mass spectrometry.

In further experiments, the uptake and use of a radioactively labeled version of methionine was followed in activated T cells. Using these approaches, Sinclair et al. showed T cell activation increased the cells demand for methionine, and that T cells need a steady supply of methionine to remain activated. The main limiting factor in this process was the speed at which the cell could make the transport systems it needs to collect methionine from its surroundings.

These findings could prove useful in developing treatments for diseases associated with uncontrollable T cells, such as leukaemia and certain autoimmune diseases. Such treatments could, for example, involve restricting the transport of methionine into T cells through drugs, or potentially via the diet.

DOI: https://doi.org/10.7554/eLife.44210.002

responses. Hence understanding how the supply of these substrates is controlled is essential. In this context, it is evident that protein and nucleotide methylations are essential for T cell differentiation: Histone and DNA methylations are key epigenetic modifications that control the accessibility of DNA to the transcriptional machinery (*Allis and Jenuwein, 2016*). The methylation of RNA is also critical for cell function: mRNA cap methylation controls mRNA binding to the eukaryotic translation initiation factor 4E (eIF4E) to regulate translation initiation (*Aregger and Cowling, 2017*; *Gonatopoulos-Pournatzis and Cowling, 2014*; *Varshney et al., 2018*); the methylation of adenosine, 'm6A', in RNA controls multiple processes including translation, splicing and stability of mRNA (*Dominissini et al., 2012*; *Louloupi et al., 2018*; *Li et al., 2017*; *Mauer et al., 2017*). Furthermore, there is an increasing awareness that protein arginine methylation is critical for T lymphocyte activation (*Inoue et al., 2018*). How do T cells 'manage' the increased demand for methyl donors as they respond to immune activation? In this context, methionine is an essential amino acid in mammals for de novo protein synthesis but it is also required for the production of S-adenosylmethionine (SAM), the universal methyl donor for DNA, RNA and protein methyltransferases. The unanswered question then is how is methionine metabolism and the production of methyl donors controlled in T cells? There are many studies showing the importance of protein, RNA, DNA and histone methyltransferases in T cells (*Inoue et al., 2018*; *Li et al., 2017*; *Lee et al., 2001*; *Thomas et al., 2012*; *Yang et al., 2015*; *Karantanos et al., 2016*; *Gray et al., 2017*; *Sellars et al., 2015*; *DuPage et al., 2015*; *Geoghegan et al., 2015*; *Zhang et al., 2014*; *Tumes et al., 2013*; *Gamper et al., 2009*). However, critical facts about methionine metabolism are unknown; do naïve T cells express all the key enzymes for methionine metabolism to provide methyl donors for methyltransferases? Is there any immune regulation of flux through the methionine cycle? It is also unknown how T cells control intracellular methionine availability in terms of the balance between the production of methionine from intracellular recycling (salvage pathways) versus external sources. How dependent is T cell activation on the external methionine supply? What are the relevant methionine transporters in T

lymphocytes and is methionine metabolism coupled to T cell activation and differentiation into effector cells?

The present study uses high resolution mass spectrometry and metabolic labelling strategies to address these fundamental questions. The data identify and quantify expression of the key enzymes of the methionine cycle in T cells and document how these are regulated by immune activation. Key observations are that methionine flux through the methionine cycle is controlled by the T cell antigen receptor and that a constant sustained external supply of methionine is necessary for T cell immune activation to sustain protein biogenesis, histone methylation and RNA methylation. We identify a dominant rate limiting step for the production of methyl donors for nucleotide and protein methylations in immune activated T cells is methionine transport across the T cell membrane. Antigen receptor regulation of methionine transport across the cell membrane is thus a key step that licenses the use of methionine for fundamental cellular process that drive T lymphocyte proliferation and differentiation.

## Results

### The sustained supply of extracellular methionine is important for activation of T cells

To investigate whether methionine availability in the external environment is important for T cells, we initially assessed the impact of methionine deprivation on the immune activation of CD4$^+$ T cells. Flow cytometric forward and side scatter analysis of CD4$^+$ T cells activated in the absence of methionine revealed that these cells do not undergo normal cell growth/blastogenesis (*Figure 1a*). TCR-activated CD4$^+$ T cells deprived of methionine are smaller than CD4$^+$ T cells activated in the presence of methionine (*Figure 1a*), and fail to proliferate (*Figure 1b*). However activation markers such as CD69 are expressed normally (*Figure 1c*). Methionine levels in RPMI tissue culture media are 100 μM. Serum methionine availability has been shown to range from 3 to 30 μM (*Mentch et al., 2015*), consequently we used levels of methionine spanning this range to further explore the importance of extracellular methionine availability for CD4$^+$ T cell differentiation. In these experiments CD4$^+$ T cells were activated by triggering TCR complexes and CD28 and cultured in Interleukin 2 (IL2) and IL12 to differentiate into Th1 cells that produce high levels of interferon gamma (IFNγ). We assessed the impact of restricting methionine availability on CD4$^+$ T cell differentiation by activating the cells in media with different levels of methionine. The data in *Figure 1d* show that the frequency of IFNγ producing CD4$^+$ T cells is dependent upon extracellular methionine availability (*Figure 1d*). A 2-fold reduction in methionine from 10 μM to 5 μM thus had a striking impact on the frequency of IFNγ producing CD4$^+$ T cells that could develop under Th1 polarising conditions (*Figure 1e*) and also controlled the amount of IFNγ produced per cell (*Figure 1f*).

These initial experiments show that methionine availability is rate limiting for T cell activation and differentiation. However, another question is how important is sustained methionine supply for T cell function? To address this question T cells were activated through TCR/CD28 in a saturating concentration of external methionine (100 μM) for 20 hr prior to culturing in reduced concentrations of methionine for a further 2 hr. We then used quantitative single cell assays to assess the impact of acute restriction of methionine availability for protein synthesis, RNA and DNA synthesis in immune activated T cells. Protein synthesis was assayed using a single cell assay that quantifies the incorporation of an analogue of puromycin into nascent protein chains in the ribosome; total RNA and DNA synthesis were measured by monitoring incorporation of the nucleoside analogue, 5-ethynyl uridine (EU) or a modified thymidine analogue (EdU) respectively. The data show that T cells activated in the presence of 100 μM methionine have high rates of protein, RNA and DNA synthesis (*Figure 1g–i*). However, limiting extracellular methionine availability for 2 hr strikingly impacted on the ability of the cells to maintain these key processes. The EC50, that is the concentration of methionine required for half maximal effect was in these TCR/CD28 activated CD4$^+$ cells was 1.39 μM for protein synthesis; 2.94 μM for RNA synthesis and 12.62 μM for the frequency of cells undergoing DNA synthesis (*Figure 1j–l*). Similar experiments were done with Th1 effector cells. These were differentiated for 5 days in 100 μM methionine and then maintained for five hours in methionine limited media. The data show that the ability of Th1 cells to sustain RNA, protein and DNA synthesis is also dependent on sustained methionine supply (*Figure 1m–o*).

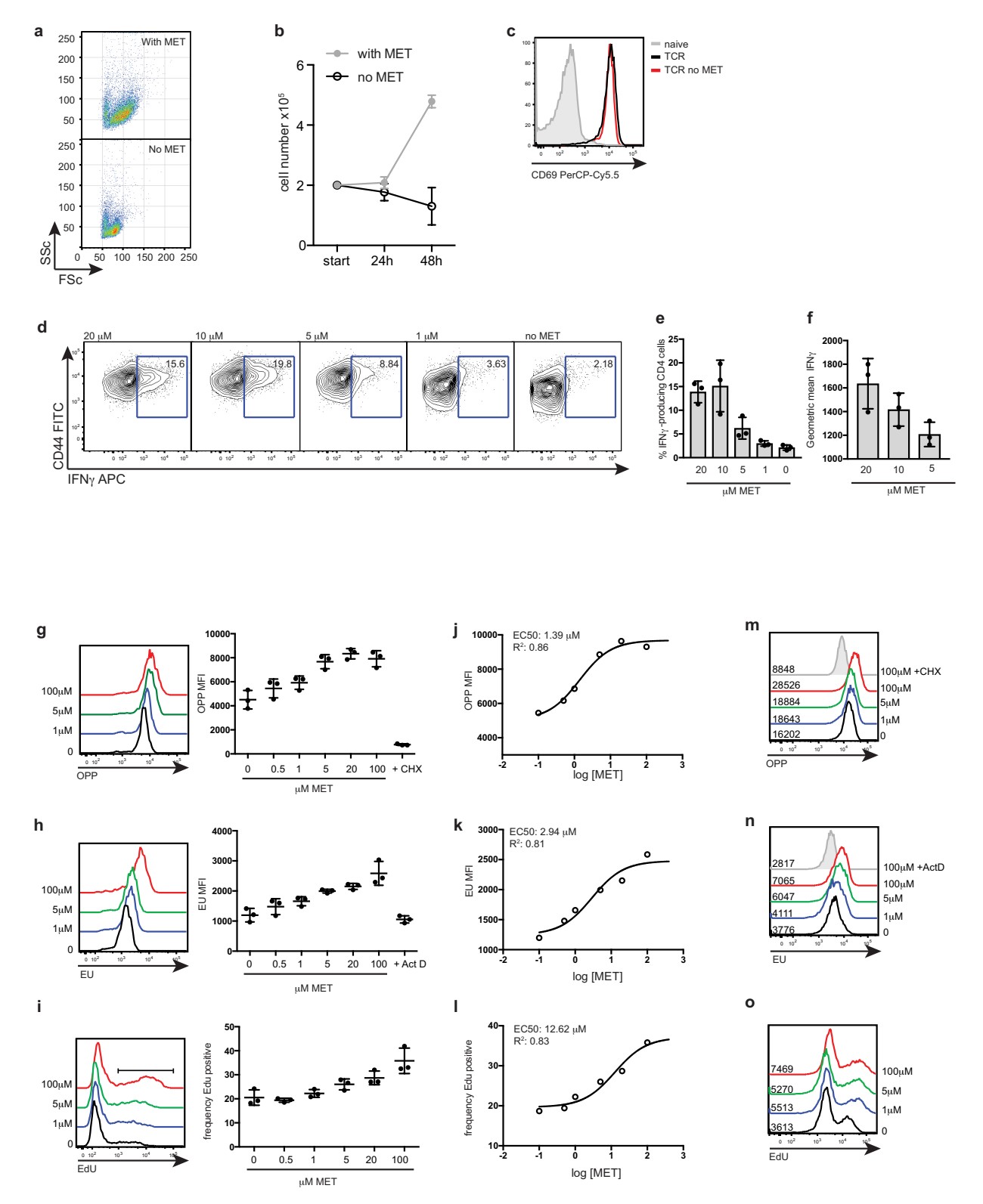

**Figure 1.** T cell activation and differentiation requires a sustained supply of extracellular methionine. (a) Flow cytometry plots show the forward (FSC) and side (SSC) scatter profiles of CD4+ T cells stimulated through the TCR (CD3/CD28) for 18 hr ± methionine. (b) Cell counts of CD4+ T cells over 48 hr after TCR-stimulation (CD3/CD28) in the presence or absence of methionine. (c) Flow cytometry plots CD69 expression of CD4+ T cells stimulated through the TCR (CD3/CD28) for 18 hr ± methionine. (d-f) CD4+ T cells activated with CD3/CD28 antibodies + IL2/IL12 for 3 days in indicated

*Figure 1 continued on next page*

*Figure 1 continued*

methionine concentrations. (**d**) CD44 surface staining and intracellular IFNγ cytokine staining. The % CD4$^+$ T cells producing IFNγ is indicated on the plot. The percentage (**e**) and MFI (**f**) of IFNγ producing CD4$^+$ T cells from three biological replicates. (**g–I**) CD4$^+$ T cells were stimulated through the TCR (CD3/CD28) for 20 hr before culturing in reducing concentrations of methionine for a further 2 hr. The histograms (left panel) show de novo protein synthesis as measured by incorporation of the puromycin analog (OPP) (**g**) or RNA synthesis as determined by EU incorporation (**h**). The right panels show the MFIs over the expanded dose response. Cyclohexamide (CHX) treatment or Actinomycin D (ActD) treatment are included as negative controls for protein and RNA synthesis. (**i**) The histograms (left panel) show the frequency of CD4$^+$ T cells undergoing DNA synthesis as measured by EdU incorporation. The right panel shows the frequency of EdU positive cells over the expanded dose response. (**j–I**) EC50 values were calculated from dose response curves (using logged concentration values). Goodness of fit is represented by the R$^2$ values. (**m–o**) Th1 effector cells were expanded for 5 days before culturing in reducing concentrations of methionine for a further 5 hr. The histograms show (**m**) de novo protein synthesis (**n**) RNA synthesis and (**o**) DNA synthesis. (Plots are representative of 3 biological replicates. Gating strategies are provided in *Supplementary file 1*. Error bars are mean ± s.d of: three biological replicates; Points on the graphs indicate biological replicates.).

DOI: https://doi.org/10.7554/eLife.44210.003

## Methionine metabolic pathways in T cells.

Methionine is the predominant 'start' amino acid used to initiate polypeptide synthesis during mRNA translation. *Figure 2a* shows naïve CD4$^+$ T cells have almost undetectable incorporation of extracellular $^3$H-methionine into protein, however incorporation of $^3$H-methionine into protein is greatly increased upon activation through the TCR (*Figure 2a*).

One explanation for the environmental methionine requirement for T cells is that it fuels protein synthesis. However methionine fuels other essential metabolic pathways, consequently we used mass spectrometry to explore methionine metabolism in CD4$^+$ T cells stimulated via the T cell antigen receptor/CD28 complex. In particular, the methionine cycle which is initiated when methionine is converted into S-adenosylmethionine (SAM) in an ATP-consuming reaction and catalysed by methionine adenosyltransferase (MAT2A). Methyltransferases then transfer the methyl group from SAM to yield S-adenosylhomocysteine (SAH) and a methylated substrate. SAH is swiftly converted into homocysteine (HCy) by S-adenosylhomocysteine hydrolase (AHCY, also known as SAHH). The T cell metabolomics data show that SAM levels remain relatively constant between TCR stimulated and naïve CD4$^+$ T cells (*Figure 2b*). However, TCR activated cells show an increase in the generation of S-adenosylhomocysteine (SAH) and HCy (*Figure 2b*). This increased production of SAH and HCy demonstrates that triggering the TCR drives increased flow through the methionine cycle. HCy has two potential metabolic fates, that is, it can be converted to cystathionine, or recycled back into methionine via subsequent enzymatic reactions through the de novo pathway. In the de novo pathway, methionine synthase (MTR) and the cofactor vitamin B12 perform the rate-limiting step of incorporating methyl groups derived from folate metabolism and HCy to produce methionine. SAM can also be utilised for polyamine synthesis, providing spermine and spermidine and yielding 5-methylthioadenosine (MTA). The sulphur of MTA can be recycled back into methionine using the *salvage pathway*, the first step of which is catalysed by MTA phosphorylase (MTAP) to generate 5-methylthioribose-1-phosphate (reviewed in *Mentch and Locasale, 2016*; *Albers, 2009*). In this context, further evidence that TCR triggering drives the methionine cycle is provided by the data showing that activated CD4$^+$ T cells accumulate 5-methylthioadenosine (MTA) and Methyl-5-thio-5-D-ribose 1-phosphate; metabolites produced downstream of polyamine synthesis. Activated CD4$^+$ T cells also had increased levels of cystathionine and 2-oxobutanoate (C4H6O3); the latter is produced when cystathionine gamma-lyase converts cystathionine to cysteine (*Figure 2b*).

One function for the methionine metabolite, S-adenosylmethionine (SAM), is as a methyl donor for methylation modification reactions. The increased production of S-adenosylhomocysteine (SAH) in TCR activated cells argues that SAM is being used as a methyl donor during T cell activation. To explore links between TCR triggering and methylation pathways we first explored if activated T cells showed increased protein methylation by monitoring the methylation status of histone H3 at both activating (lysine 4; H3K4) and inhibitory (lysine 27; H3K27) sites. To investigate the methylation status of H3K4 and H3K27 in CD4$^+$ T cells we used flow cytometry with antibodies that recognise trimethylation on H3K4 and H3K27, respectively. These experiments show that trimethylation (me3) of both H3K4 and H3K27 in CD4$^+$ T cells is increased upon in vitro TCR- stimulation (*Figure 2c*). *Figure 2d* shows the ratio of staining of total H3, H3K27me3 and H3K4me3 of TCR-stimulated, compared with non-stimulated CD4$^+$ T cells (*Figure 2d*). We also addressed whether a similar increase in

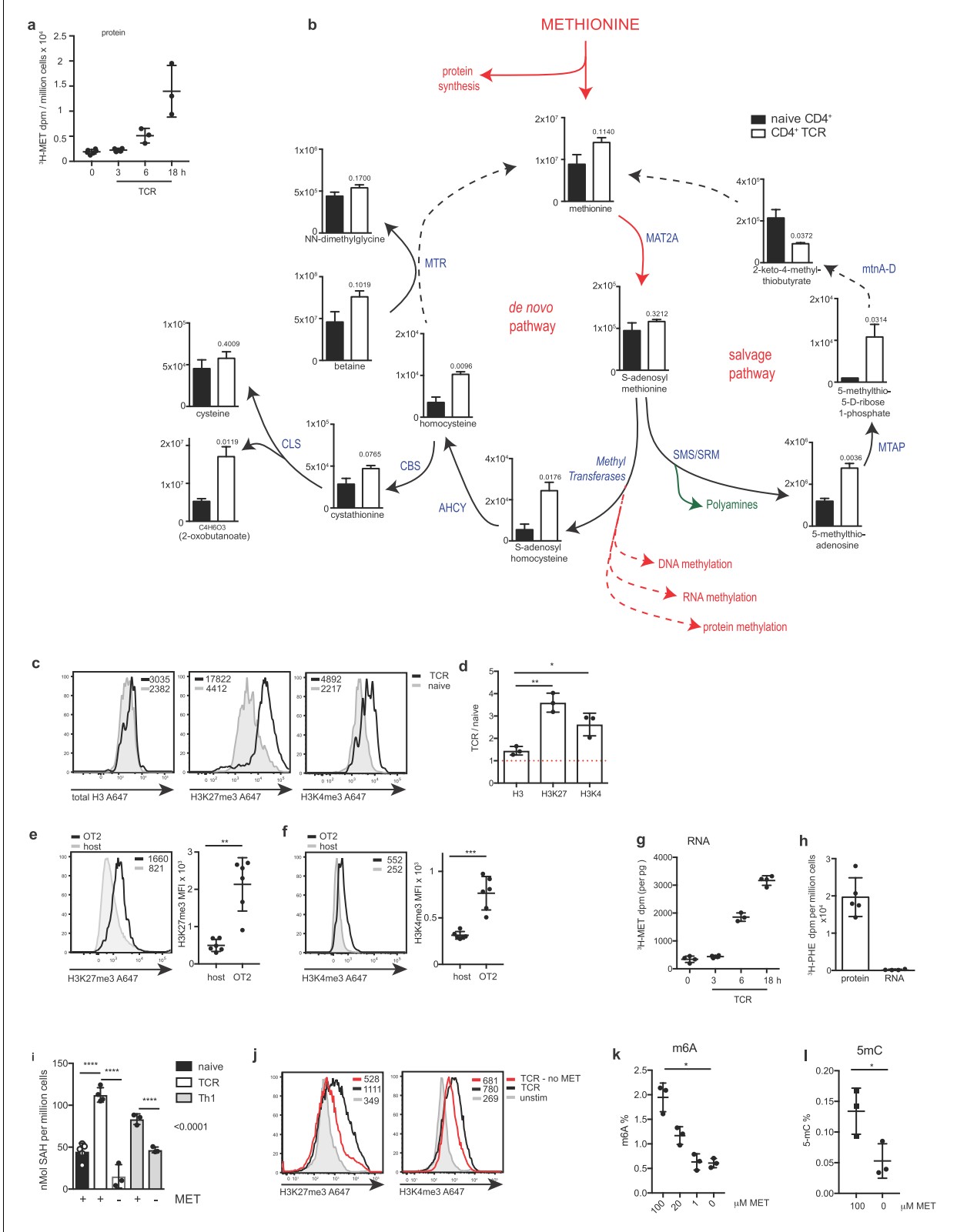

**Figure 2.** Methionine metabolism in T cells. (**a**) [3]H radioactivity measured in TCA precipitated protein from isolated CD4[+] T cells stimulated through the TCR (CD3/CD28) in the presence of [3]H-methionine for the indicated times. (**b**) Metabolomic analysis of metabolites in the de novo pathway and the salvage pathway of the methionine cycle. The graphs show metabolite intensity derived from integrated peak areas of MS intensity from naïve CD4[+] T cells and TCR-stimulated CD4[+] T cells (CD3/CD28, for 16 hr). Enzymes are indicated adjacent to arrows (in blue). P values are indicated on each graph.

*Figure 2 continued on next page*

*Figure 2 continued*

Source data is available in *Figure 2—source data 1*. (c) The histograms show representative intracellular staining of total H3, trimethylated H3K27 (H3K27me3) or trimethylated H3K4 (H3K4me3) from IL7 maintained (unstimulated) or TCR-stimulated (CD3/CD28) CD4$^+$ T cells for 18 hr. Geometric mean fluorescence intensities (MFI) are shown in the histograms. (d) The graph shows ratios of H3, H3K27me3 and H3K4me3 MFIs from TCR-stimulated (CD3/CD28) CD4$^+$ T cells compared to unstimulated CD4$^+$ T cells. (e, f) OT2 (CD45.1) cells were adoptively transferred into WT CD45.2 hosts. The hosts were immunised with NP-OVA/alum and the transferred OT2 cells were analysed after 3 days. The histograms (left) show representative intracellular staining of H3K27me3 (e) and H3K4me3 (f). Graphs (right) show ratios of H3K27me3 and H3K4me3 MFIs in activated OT2 CD4$^+$ T cells compared to non-activated host CD4$^+$ T cells 3 days post-immunisation. (g) $^3$H radioactivity measured in RNA extracted from isolated CD4$^+$ T cells stimulated through the TCR (CD3/CD28) in the presence of $^3$H-methionine for the indicated times. (h) $^3$H radioactivity measured in protein or RNA extracted from CD4$^+$ T cells stimulated in parallel to (g) in the presence $^3$H-phenylalanine for 18 hr. (i) Levels of SAH from unstimulated (naïve cells), TCR-stimulated (CD3/CD28, 18 hr) CD4$^+$ T cells or IL2 maintained Th1 cells ± methionine for 18 hr. (j) The histograms show representative intracellular staining of total H3, trimethylated H3K27 (H3K27me3) or trimethylated H3K4 (H3K4me3) from IL7 maintained (unstimulated) or TCR-stimulated (CD3/CD28) CD4$^+$ T cells±methionine as indicated (18 hr). (k) Percentage of RNA with m6A modification, as determined by ELISA, in Th1 cells cultured with decreasing methionine concentrations for 5 hr (as indicated). (l) Percentage of RNA with m5C modification, as determined by ELISA, in Th1 cells cultured ±methionine for 5 hr. (Error bars are mean ± s.d of: (a–d, i–l) three biological replicates (e–f) six biological replicates. (g,h) 4 (RNA) biological replicates and five biological replicates (protein). MFIs are indicated in the histograms, points on the graphs indicate biological replicates. (b, e, l) *t*-test; (d,i,k) One-way ANOVA; *P*= *<0.05, **<0.01, ***<0.001, ****<0.0001; Flow cytometry gating strategies are provided in *Supplementary file 1*).
DOI: https://doi.org/10.7554/eLife.44210.004
The following source data is available for figure 2:

**Source data 1.** Spreadsheet containing the list of metabolite intensities derived from integrated peak areas of MS intensity from naïve CD4$^+$ T cells (N1-3) and TCR-stimulated CD4$^+$ T cells (S1-3).
DOI: https://doi.org/10.7554/eLife.44210.005

H3K27me3 and H3K4me3 occurs in vivo. Accordingly, CD4$^+$ T cells from OT2 TCR transgenic mice were adoptively transferred into normal hosts prior to immunisation with the cognate antigen, ovalbumin. Analysis of the H3K27me3 and H3K4me3 staining shows that H3 trimethylation on K27 and K4 increases upon immune activation in vivo (*Figure 2e,f*).

SAM is also used as a substrate for RNA methyltransferases, producing methylated RNA and SAH. To investigate if there are changes in RNA methylation during T cell activation we designed experiments where CD4$^+$ T cells were activated in methionine-free media supplemented with $^3$H-methionine where the radiolabel is attached to the methyl group. This allowed us to purify RNA from the activated cells and quantified the incorporation of the $^3$H-methyl group into RNA. The data show that the $^3$H-radiolabel is incorporated into RNA after 6 hr of TCR/CD28 stimulation, and this is further increased after 18 hr (*Figure 2g*). A key control in this experiment is to ensure that there is no protein contamination in the RNA extractions. For this we did a parallel quantification of the incorporation of $^3$H-methionine and $^3$H-phenylalanine: the latter would only be present into cellular proteins. The data show there was no protein contamination of these RNA preparations as judged by absence of any detectable $^3$H-phenylalanine label, compared with high levels of $^3$H-phenylalanine incorporated into protein (*Figure 2h*).

How dependent are these protein and nucleotide methylation reactions on extracellular methionine? *Figure 2i* shows that SAH levels in TCR activated CD4$^+$ T cells and Th1 cells are dependent upon extracellular methionine supply which argues that sustained methionine availability is required to produce methyl donors for protein and nucleotide methylation (*Figure 2i*). In this context, the data in *Figure 2j* show that the increase in H3K4 and H3K27 trimethylation in CD4$^+$ T cells in response to TCR activation is regulated by extracellular methionine availability (*Figure 2j*). We also examined the importance of extracellular methionine for RNA methylation in T cells. One of the most abundant mammalian RNA modifications is methylation of the N6 adenosine (m6A) regulated by the METTL3 methyltransferase (*Liu et al., 2014a*). This RNA methylation is important for mRNA stability and in T cells has been shown to be essential for normal T cell function (*Li et al., 2017*). The data in *Figure 2k* show that the percentage of total m6A methylation in effector Th1 cells decreases as extracellular methionine is decreased (*Figure 2k*). Another RNA post-translational methyl modification found in RNA is 5-methylcytosine (m5C), the data in *Figure 2l* show that the amount of m5C in mRNA from effector Th1 cells is dependent upon extracellular methionine (*Figure 2l*). Collectively, these data demonstrate that extracellular methionine is not only directed into protein synthesis in T cells but also into the methionine cycle to generate methyl donors for histone and RNA methylation

reactions. Furthermore, the sustained supply of extracellular methionine is necessary for these processes.

## Methionine cycle regulation in T cells

The elevated levels of SAH and HCy and the increased levels of RNA and histone methylation demonstrate that immune activated T cells increase metabolic flow through the methionine cycle. To explore the molecular basis for the increases in methionine metabolism in activated T cells we used quantitative, mass spectrometry-based proteomics to interrogate the expression and abundance levels of methionine cycle enzymes in naïve, TCR activated and effector CD4$^+$ T cells. Critical enzymes that regulate the metabolic flow through the two arms of the methionine cycle, the de novo and salvage pathways, are shown in *Figure 3a,b*. We initially looked at the expression levels of these key proteins in naïve T cells by comparing their relative abundance against the backdrop of the total protein landscape. The data show that naïve CD4$^+$ T cells express the key methionine cycle enzymes MAT2A, AHCY and MTR (*Figure 3c,d and e*). Interestingly, both MAT2A and AHCY are highly abundant proteins in the naïve cell proteomic landscape (*Figure 3c and d*). Similarly, SRM/SMS and MTAP, enzymes which use SAM for polyamine synthesis and subsequent methionine salvage, are abundantly expressed in naïve T cells (*Figure 3f and g*). mtnA-D, enzymes involved in the final steps of methionine salvage, are also expressed at levels higher than the 'average' protein is expressed in a T cell (*Figure 3h*). The relative abundance, as indicated by their concentration, of these proteins within the total proteomic landscape is consistently maintained at high levels upon activation and differentiation of CD4$^+$ T cells (*Figure 3i*).

The data in *Figure 2* show that immune activation of T cells drives increased RNA methylation. We therefore interrogated the proteomic data to quantify expression of essential RNA methyltransferases in naïve and immune activated T cells. mRNA methyl cap formation marks RNAs for further processing, nuclear export and translation initiation (*Gonatopoulos-Pournatzis and Cowling, 2014*); the m6A RNA methylation regulates mRNA stability (*Geula et al., 2015*). Key methyltransferases for RNA cap methylation are RNMT, the RNA (guanine-7-) methyltransferase (*Cowling, 2009*), and the cap methyl transferases CMTR1 or CMTR2 (*Inesta-Vaquera and Cowling, 2017*). The proteomics data show that RNMT and CMTR1 are abundantly expressed in naïve and at similar levels in TCR activated CD4$^+$ T cells, whereas CMTR2 is not expressed at high levels in any population (*Figure 4a*). The data show that naïve T cells also express equimolar levels of the METTL3/METTL14 m6A methyltransferase complex (*Figure 4b*) (*Liu et al., 2014b*). Another frequent RNA modification is cytosine $-5$ methylation (m5C). The proteomics data show that naïve T cells express high levels of NSUN2, the methyltransferase responsible for RNA m5C modification, these high levels are maintained after TCR activation and through differentiation (*Figure 4c*) (*Yang et al., 2017*).

The high levels of expression of the RNA methyl transferases in naïve T cells is in contrast to the pattern of expression of the histone and DNA methyl transferases. The expression of the DNA methyl transferases DNMT1 and DNMT3a is markedly higher in immune activated CD4$^+$ T cells than in naïve T cells (*Figure 4d*). Moreover, activated CD4$^+$ T cells dramatically increase expression of UHRF1, which is required for DNMT1 recruitment and activity (*Nishiyama et al., 2016*). The expression of the polycomb repressive complex 2 (PRC2), which is responsible for histone 3 lysine 27 methylation (H3K27), is also highly increased upon T cell activation and in differentiated effector T cells compared with naïve T cells (*Figure 4e,f*) as is expression of other key histone methyltransferases, responsible for histone methylations including H3K4, K9 and K36 and H4K20 methylation (*Bochyńska et al., 2018*; *Mozzetta et al., 2015*; *Wagner and Carpenter, 2012*) (*Figure 4g*).

One striking observation in the current study was that even short periods of methionine deprivation cause a loss of RNA and histone methylations. A key question then, is whether the dependence of these cells on extracellular methionine for histone and RNA methylation reflects that methionine is needed to sustain expression of the methionine cycle enzymes and/or RNA and histone methyltransferases. To address this question, we used mass spectrometry to assess the impact of acute methionine deprivation on the proteome of effector Th1 CD4$^+$ T cells. In these experiments, Th1 CD4$^+$ T cells were switched into media containing 1µM or 100µM methionine for 5 hr prior to processing for single-shot proteomics from which we identified 4'400 proteins. The data shows that the protein expression profile did not significantly vary upon acute methionine deprivation, with a very high correlation coefficient between the two conditions of 0.98 (*Figure 5a*). Proteins that showed significant changes in expression (increased or decreased) are listed in *Figure 5b*.

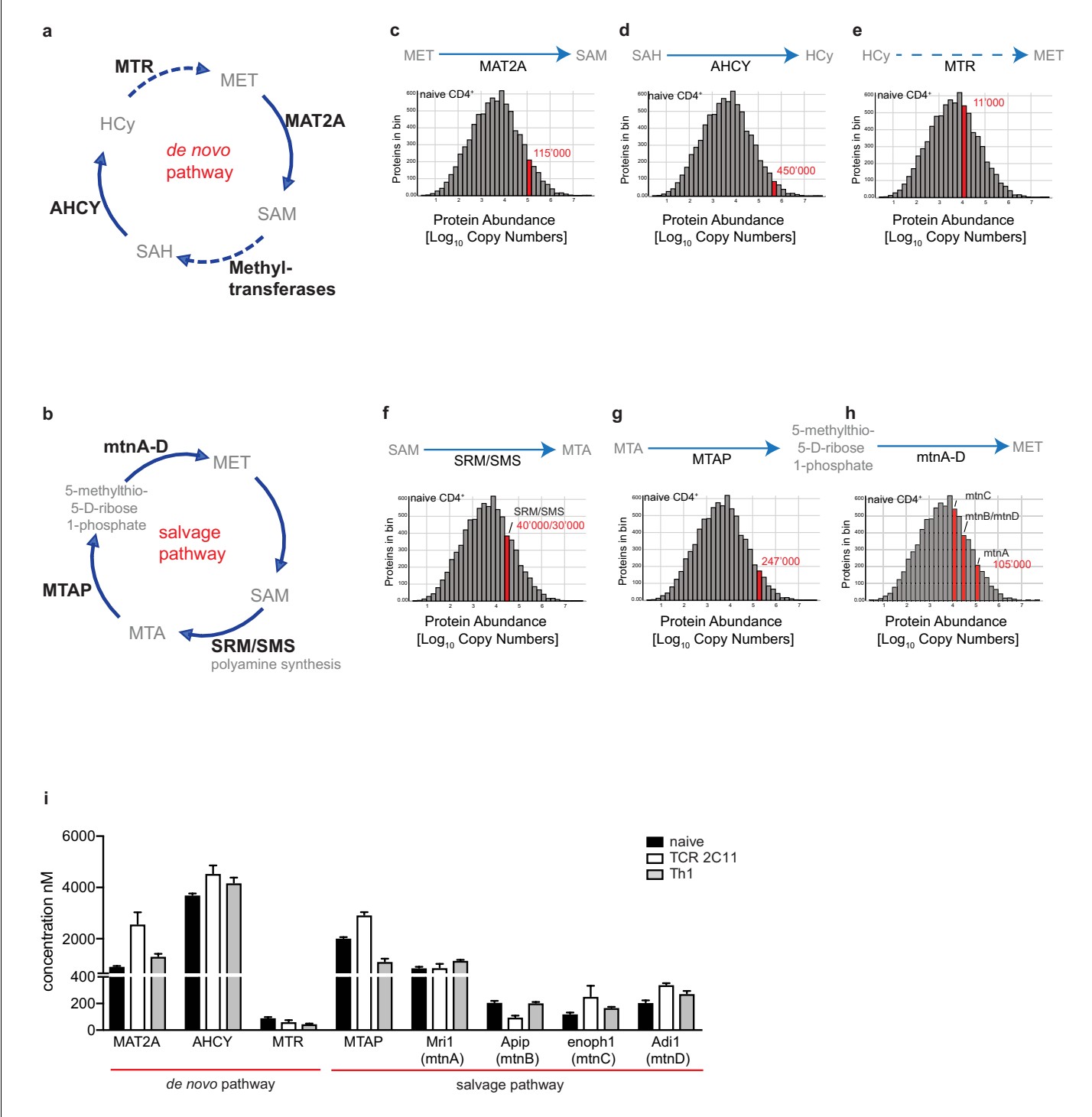

**Figure 3.** Proteomics expression of methionine pathway in T cells Quantitative proteomics data showing the abundance of key enzymes in the methionine cycle of the de novo. (**a**) and the salvage pathways (**b**). Mean protein copy numbers, indicated in red, estimated using the proteomic ruler protocol and presented as log-transformed mean values are shown relative to their frequency in the total data set. The graphs show copy numbers from naïve CD4$^+$ T cells of MAT2A (**c**), AHCY (**d**), MTR (**e**) and SRM/SMS (**f**), MTAP (**g**) and mtnA-D (**h**). (**i**) Quantitative proteomics data comparing the concentration of key enzymes in the methionine cycle of the de novo and the salvage pathways in naïve, TCR activated and Th1 effector CD4$^+$ T cells. Concentration is calculated using the histone ruler and estimated mass of molecules per cell, as described in **Wiśniewski et al. (2014)**. (Data are from three biological replicates and error bars are mean ± s.d.).

DOI: https://doi.org/10.7554/eLife.44210.006

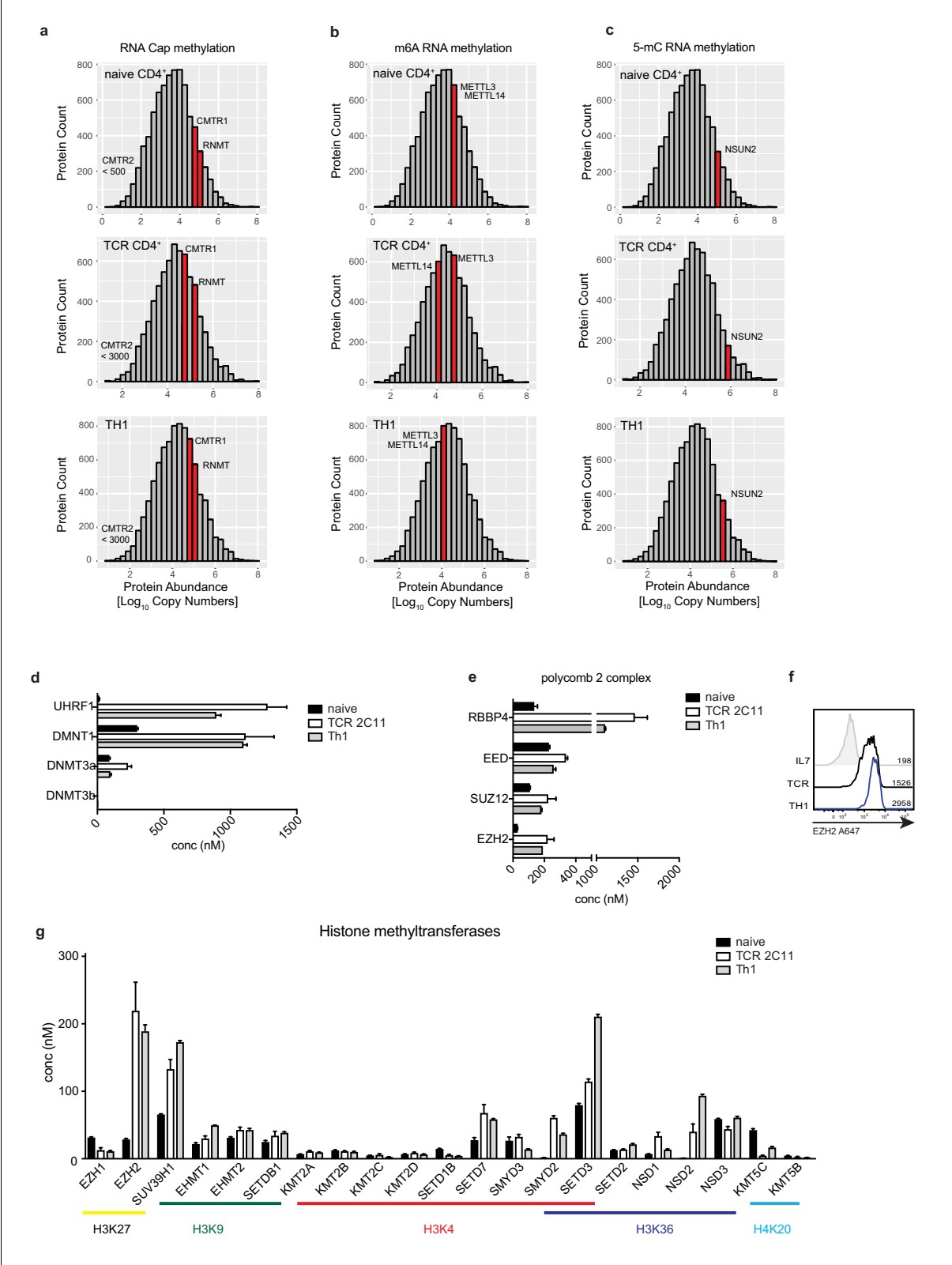

**Figure 4.** Methyltransferase expression in T cells. Abundance of (**a**) CMTR1 and RNMT cap methyltransferases (**b**) METTL3 m6A RNA methyltransferase and (**c**) NSUN2 m5C methyltransferase expression in naïve, TCR activated and effector Th1 CD4[+] T cells (plotted as in 3 c). (**d**) DNA methyltransferases use SAM as a methyl donor to methylate CG residues. The concentration of the DNA methyltransferase complexes expressed in naïve CD4[+] T cells (naïve), 24 hr TCR- stimulated (aCD3/aCD28, IL2/12) CD4[+] T cells (TCR) and in vitro generated Th1 cells (Th1). (**e**) The polycomb repressor complex 2

*Figure 4 continued on next page*

*Figure 4 continued*

(PRC2) uses SAM to methylate lysine residues for example K27 on histone tails. Concentration of the PRC2 components calculated from proteomics data from naïve CD4$^+$ T cells (naïve), 24 hr TCR- stimulated CD4$^+$ T cells (TCR) and in vitro generated Th1 cells (Th1). (f) Flow cytometry plots showing EZH2 staining in naïve CD4$^+$ T cells, CD4$^+$ T cells stimulated through the TCR (aCD3/aCD28, IL2/12) for 18 hr and in vitro generated Th1 cells. MFI are shown on the graph. (g) Concentrations of histone methyltransferases calculated from proteomics data of naïve CD4$^+$ T cells (naïve), 24 hr TCR-stimulated CD4$^+$ T cells (TCR) and in vitro generated Th1 cells (Th1). Histone methyl modifications are indicated. (Error bars are mean ± s.d. Data are from three biological replicates.).

DOI: https://doi.org/10.7554/eLife.44210.007

With regard to the methionine cycle specifically, these experiments found that acute short term methionine deprivation of Th1 cells had no impact on expression of key enzymes including; MAT2A, AHCY, SRM/SMS,MTAP and mtnA (*Figure 5c–g*). The expression of RNA, DNA and histone methyl transferases was also not changed by short-term methionine restriction (*Figure 5h–j*). The need for T cells to sustain supply of methionine to maintain the methionine cycle is thus explained by the need for methionine to produce methyl donors.

## The impact of methionine restriction on c-myc expression and mTORC1 activity

How much does methionine deprivation impact on other metabolic programs in T cells? In this context, the serine/threonine kinase mTORC1 and the transcription factor c-myc can be described as hubs of metabolic regulation in T cells. Appropriate mTORC1 activity and c-myc expression are critical for correct effector T cell responses (*Xu et al., 2012*; *Zeng and Chi, 2017*; *Wang et al., 2011*). In both T cells and NK cells, expression of c-myc protein is highly sensitive to amino acid availability and expression of amino acid transporters (*Sinclair et al., 2013*; *Loftus et al., 2018*). To explore whether c-myc expression was dependent upon extracellular methionine availability, we used a mouse model where a fusion protein of GFP-Myc is expressed under the control of the endogenous Myc promoter (GFP-Myc$^{KI}$) (*Huang et al., 2008*; *Preston et al., 2015*). GFP-Myc$^{KI}$ CD4$^+$ T cells increase expression of the activation marker CD69 in response to TCR/CD28 stimulation in both the presence (100µM) and absence of methionine. The data show that the induction of GFP-Myc expression in response to TCR/CD28 activation is blunted in the absence of methionine (*Figure 6a,b*).

It has also recently been demonstrated that methionine availability regulates mTORC1 activity through metabolite SAM binding to SAMTOR; SAM- SAMTOR then associates with and inhibits GATOR1, resulting in lysosomal recruitment and activation of mTORC1 (*Gu et al., 2017*; *Valvezan and Manning, 2019*). To investigate whether mTORC1 activity in T cells is sensitive to extracellular methionine availability, effector CD4$^+$ Th1 cells were switched to methionine free conditions and the impact of this on mTORC1 activity in Th1 cells was assessed. The data show mTORC1 activity in T cells was partially sensitive to methionine deprivation, though the impact of methionine loss was not equivalent to total amino acid deprivation or rapamycin treatment (*Figure 6c,d*). Treatment with SAM (50µM) modestly restores mTORC1 activity in the absence of methionine (*Figure 6e, f*). These data show that methionine availability and the production of SAM can feed into control of mTORC1 activity. The impact of methione deprivation on c-myc expression and mTORC1 activity highlights the diversity of cellular responses in T cells that are able to sense the external methionine environment.

## Control of the methionine cycle in T cells via controlled expression of methionine transport

The proteomic data reveal that increased flux through the methionine cycle and increases in RNA methylation are not explained by changes in the abundance of the key enzymes of the methionine cycle or the RNA methyl transferases. However, the sustained availability of extracellular methionine is critical which raises the possibility that the rate limiting step for methionine metabolism in activated T cells is the rate of methionine transport. To investigate the methionine transport capacity of naïve versus effector T cell populations, we compared radiolabeled methionine uptake in naïve and activated T lymphocytes. Naïve CD4$^+$ T cells show very little/near undetectable uptake of $^3$H-labeled methionine (*Figure 7a*) whereas methionine transport was readily detected in CD4$^+$ T cells activated with CD3/CD28 crosslinking antibodies and in effector Th1 cells (*Figure 7a*). *Figure 7b* show that

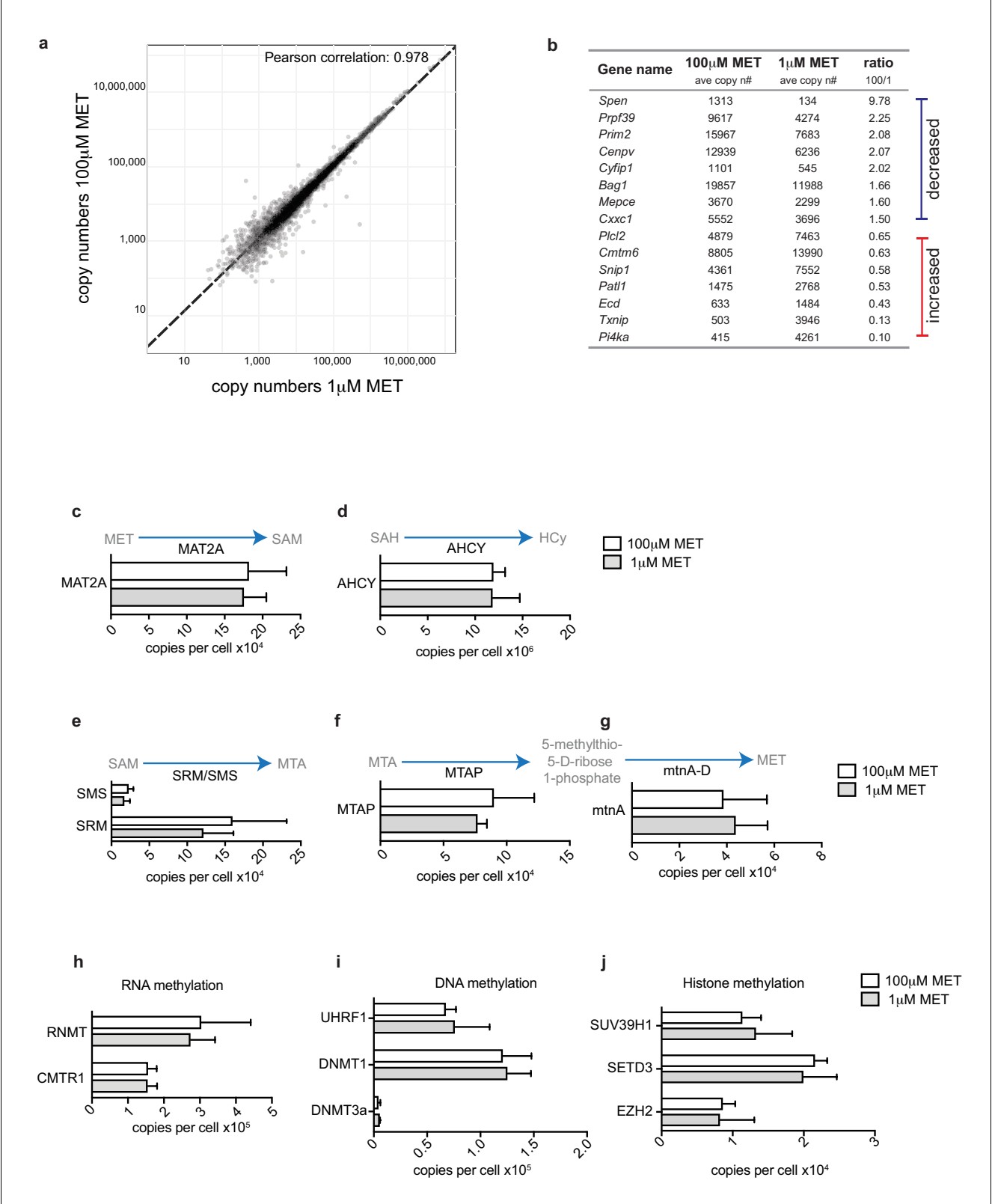

**Figure 5.** Acute methionine restriction on methionine cycle proteome Quantitative 'single-shot' proteomics was performed on in vitro generated IL2 maintained Th1 cells cultured for 5 hr in 100 μM or 1 μM methionine. (a) The graph shows the protein copy numbers in Th1 cultured with 1 μM methionine plotted against those in 100 μM methionine. Pearson correlation is indicated. (b) Proteins that were significantly differentially expressed (*P* = <0.05, with a 1.5 fold cut-off) are listed by gene name. They are ranked as most decreased by acute methionine deprivation to those that are

*Figure 5 continued*

increased. (c-g) The graphs show mean copy numbers (estimated using the proteomic ruler protocol) of (c) MAT2A, (d) AHCY, (e) SMS/SRM, (f) MTAP and (g) mtnA. (h-j) The graphs show mean copy numbers of (h) RNA cap methyltransferases RNMT and CMTR1; (i) DNA methyltransferase complex components UHRF1, DNMT1 and DNMT3a and (j) histone methyltransferases SUV39H1, SETD3 and EZH2. (Error bars are mean ± s.d. Data are from three biological replicates.).

DOI: https://doi.org/10.7554/eLife.44210.008

the high levels of methionine transport in Th1 cells is dependent on sustained signalling via the IL2 receptor. Hence, either the removal of IL2, or the exposure of cells to limiting IL2 concentrations, results in a decrease of [3]H-methionine uptake. Together, these data demonstrate that naïve CD4[+] T lymphocytes greatly increase methionine uptake in response to antigenic stimulation. These changes in methionine transport could reflect changes in either the expression or activity of T cell methionine transporters.

Mammalian methionine transporters include the System ASC (alanine-serine-cysteine preferring) transporter SLC1A5; the System L transporters SLC7A5 and SLC7A8; the System y + L transporters SLC7A6 and SLC7A7; and the System A transporters SLC38A1 and SLC38A2 (*Utsunomiya-Tate et al., 1996*; *Baird et al., 2009*; *Bröer and Palacín, 2011*; *Kanai et al., 1998*; *Nii et al., 2001*; *Napolitano et al., 2015*). Interrogation of the naïve, TCR and effector T cell proteomic data identified several candidate methionine transporters in TCR stimulated CD4[+] T cells and effector Th1s; notably SLC1A5, SLC7A5, SLC7A6 and SLC38A2 (SNAT2) (*Figure 7c*). The candidate methionine

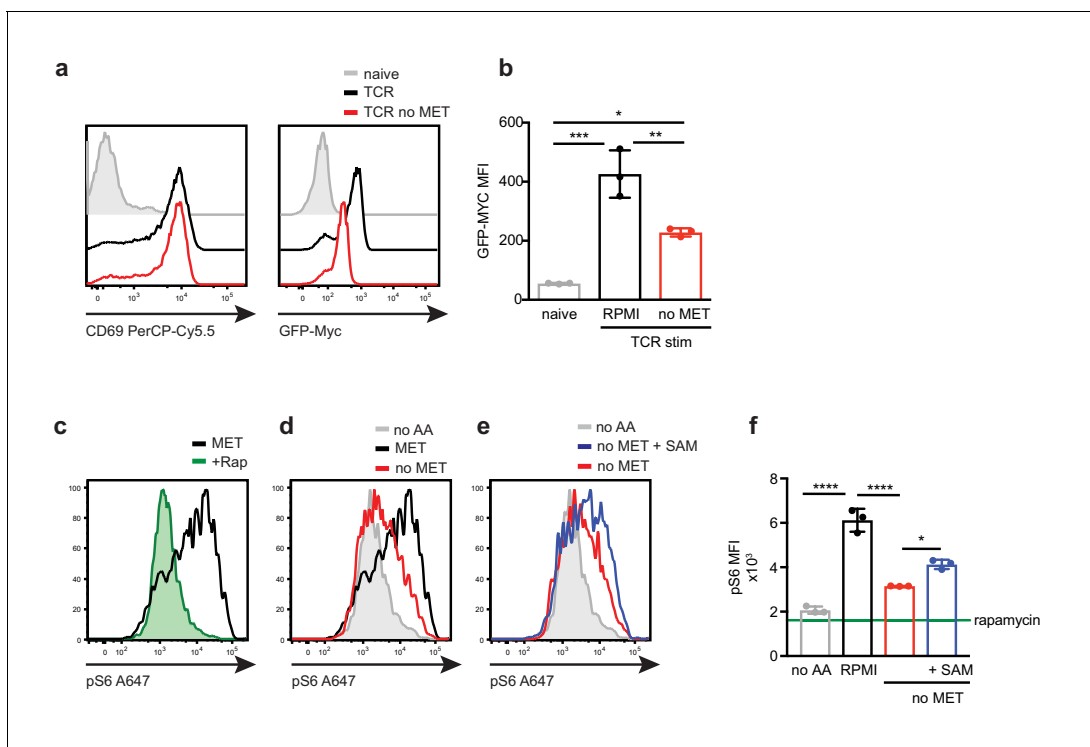

**Figure 6.** Impact of methionine restriction on c-myc expression and mTORC1 activity. (a, b) CD4[+] T cells from GFP-Myc[KI] mice were activated with CD3/CD28 antibodies ± methionine for 5 hr. (a) Flow cytometry plots show CD69 expression (left) and GFP-Myc expression (right) of IL7 maintained control CD4[+] T cells (grey), CD4[+] T cells activated with (black) or without (red) methionine. The MFI are shown in (b). (c-e) In vitro generated Th1 CD4[+] T cells were cultured for 3 days prior to acute methionine deprivation (1 hr). (c) Histograms show ribosomal protein S6 phosphorylation (pS6) in Th1 cells in methionine replete media (100 μM, MET)±rapamycin (Rap, 20 nM); (d) methionine free media (no MET), no amino acids (no AA) (e) or methionine free media supplemented with SAM (100 μM, no MET +SAM). The corresponding MFI are shown in (f). (a, c-e Data are representative of 3 biological replicates. (b,f) Points indicate individual biological replicates. Error bars are mean ± s.d. One-way ANOVA; (d) *t-test*; P = *<0.05, **<0.01, ***<0.001, ****<0.0001).

DOI: https://doi.org/10.7554/eLife.44210.009

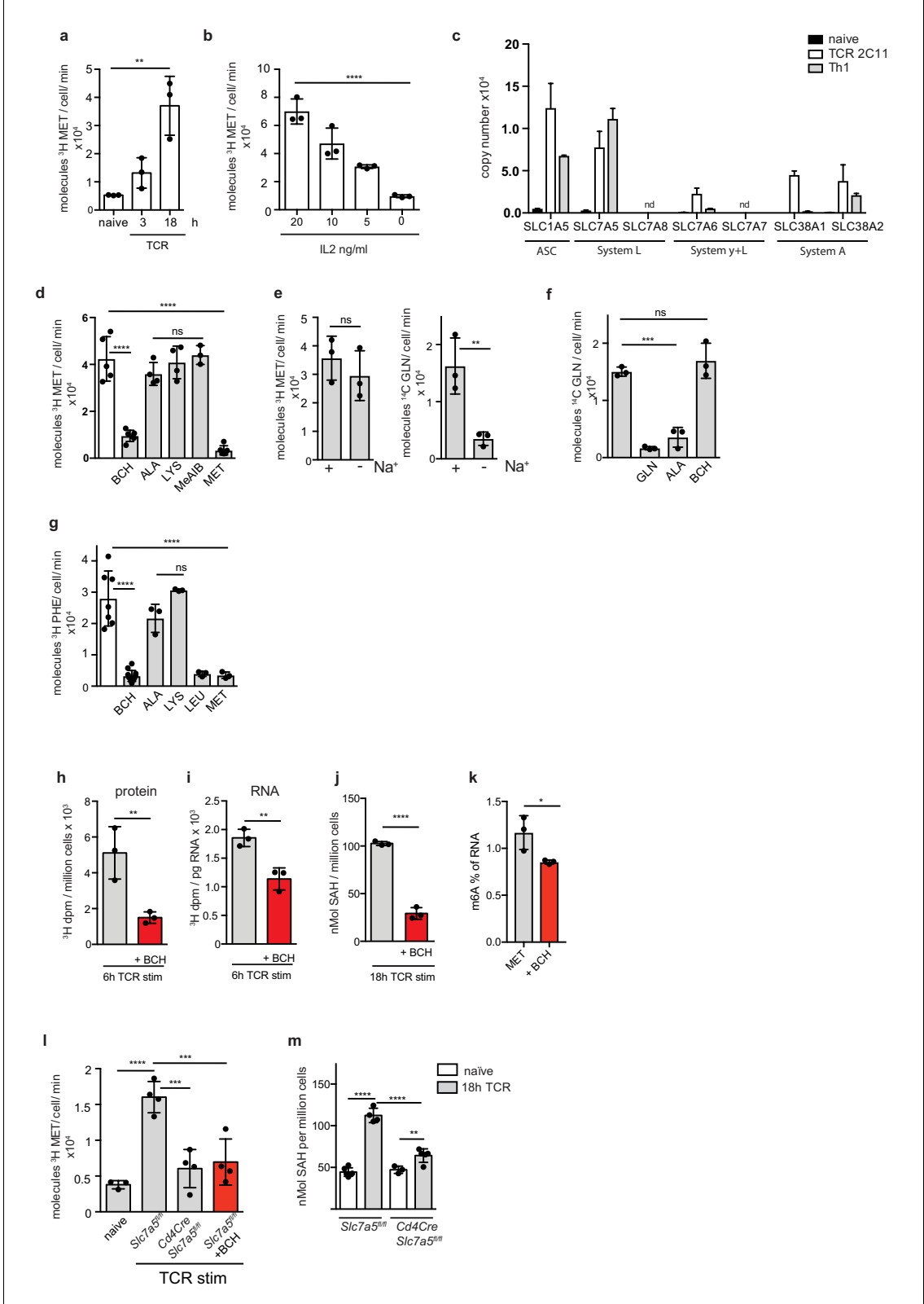

**Figure 7.** Antigen receptor and cytokine signalling regulate methionine bioavailability through SLC7A5 expression. (a) Uptake of [3]H-methionine in purified CD4[+] T cells±TCR activation using CD3/CD28 antibodies for 3 or 18 hr. (b) [3]H-methionine uptake in 5 day in vitro expanded Th1 cells switched for final 18 hr into indicated concentrations of IL2. (c) The graphs show copy numbers of potential methionine transporters from proteomics data sets of naïve CD4[+] T cells, 24 hr TCR- stimulated CD4[+] T cells and effector Th1 cells. (nd = not detected) (d) Uptake of [3]H-methionine in IL2 maintained Th1

*Figure 7 continued on next page*

Figure 7 continued

cells in the presence or absence of BCH, ALA, LYS, MeAIB or MET (all 5 mM). (e) Uptake of $^3$H-methionine (left panel) or $^{14}$C glutamine (right panel) in IL2 maintained Th1 cells in presence or absence of sodium in the uptake buffer. (f) $^{14}$C-glutamine uptake in IL2 maintained Th1 cells in the presence or absence of GLN, ALA and BCH (all 5 mM). (g) $^3$H-phenylalanine uptake in IL2 maintained Th1 cells in the presence or absence of BCH, ALA, LYS, LEU or MET (all 5 mM). (h–i) $^3$H radioactivity of TCA precipitated protein (h) or RNA (i) from CD4$^+$ T cells stimulated through the TCR (CD3/CD28) for 6 hr in the presence of $^3$H-methionine ± the System L inhibitor BCH. (j) SAH levels as determined by ELISA in CD4$^+$ T cells stimulated through the TCR (CD3/CD28)±the System L inhibitor BCH for 18 hr. (k) Percentage of RNA with m6A modification, as determined by ELISA, in Th1 cells cultured in 20 µM MET ±BCH for 5 hr. (l) Uptake of $^3$H-methionine in TCR stimulated (CD3/CD28, 18 hr) CD4$^+$ T cells from $Slc7a5^{fl/fl}$ or $Cd4$-$Cre^+$::$Slc7a5^{fl/fl}$ mice, compared to unstimulated CD4$^+$ T cells maintained in IL7,±System L transporter inhibitor BCH (5 mM). (m) SAH levels in CD4$^+$ T cells from $Slc7a5^{fl/fl}$ or $Cd4$-$Cre^+$::$Slc7a5^{fl/fl}$ mice stimulated through the TCR (CD3/CD28) for 18 hr, compared to naive cells. ((a,b- d, f, g, l, m) ANOVA, (e,h–k) $t$-test; P= *<0.05, **<0.01, ***<0.001, ****<0.0001. Uptakes performed in triplicate. Error bars are s.d. from minimum three biological replicates. Points indicate individual biological replicates.).

DOI: https://doi.org/10.7554/eLife.44210.010

transporters that were detected in activated T cells have different expression levels: the most abundant candidates were SLC7A5 and SLC1A5 (ASCT2). SLC7A6, and SLC38A2 (SNAT2) are both expressed at far lower levels than SLC7A5. The proteomic data moreover reveal the basis for the failure of naïve T cells to transport methionine: they have a very low copy number of any candidate methionine transporter (*Figure 7c*). Hence in contrast to methionine cycle enzymes which are abundantly expressed in naïve and effector CD4$^+$ T cells, expression of methionine transporters is restricted to TCR activated and effector T cells.

One way to address which methionine transporter dominates in activated T cells is to use selective pharmacological approaches that would distinguish the different candidates. For example, 2-aminobicyclo-(2,2,1)-heptane-2-carboxylic acid (BCH) is a competitive blocker for System L transporters (*Verrey et al., 2004*); MeAIB competitively blocks SLC38A2 (*Mackenzie and Erickson, 2004*). Alanine has a high affinity for SLC1A5 (*Kanai and Hediger, 2004*); lysine is transported preferentially by SLC7A6. Accordingly, alanine and lysine competition of methionine transport can be used as evidence for involvement of SLC1A5 or SLC7A6 respectively (*Verrey et al., 2004*). Furthermore, SLC1A5, SLC38A2 and SLC7A6 are all dependent on sodium to mediate methionine transport (*Kanai and Hediger, 2004*; *Mackenzie and Erickson, 2004*; *Verrey et al., 2004*). Accordingly we assessed the biochemical properties of the methionine transporters in activated T cells. The data show that $^3$H-methionine uptake in CD4$^+$ effector Th1 cells is blocked by BCH. The data also show that competition by either Alanine, Lysine or MeAIB which would block System A mediated uptake has little impact on methionine transport in activated CD4$^+$ T cells (*Figure 7d*). Furthermore, $^3$H-methionine uptake in effector CD4$^+$ T cells is sodium independent (*Figure 7e*). This contrasts with glutamine uptake in the same effector cells, which is not affected by BCH and is sodium dependent (*Figure 7e,f*). Collectively these data establish that methionine transport in activated T cells is via System L transporters. Further evidence for this conclusion is shown in *Figure 7g-k*: the cellular uptake of the System L substrate $^3$H-phenylalanine is competitively inhibited by methionine (*Figure 7g*); $^3$H-methionine incorporation into protein in activated T cells is blocked by the System L transport inhibitor BCH (*Figure 7h*); TCR induced production of methylated RNA is prevented in the presence of BCH (*Figure 7i*); BCH treatment reduces SAH levels in TCR activated CD4$^+$ T cells (*Figure 7j*); m6A mRNA methylation is reduced in the presence of BCH (*Figure 7k*). Collectively these data show that methionine delivery mediated via System L amino acid transporters is the rate limiting step for the methionine cycle in T cells and rate limiting for protein synthesis and RNA methylation. The only System L candidate identified in the proteomics was SLC7A5 and this was also very abundant (*Figure 7c*). To test the involvement of SLC7A5 directly we examined the methionine transport capacity of T cells lacking SLC7A5 expression. *Figure 7l* shows that SLC7A5 null CD4$^+$ T cells do not increase methionine uptake in response to activation with CD3/CD28 antibodies. SLC7A5 null CD4$^+$ T cells T cells also show a striking decrease in their ability to increase levels of the methionine metabolite SAH in response to TCR/CD28 stimulation (*Figure 7m*). This is consistent with SLC7A5 expression being required for the increased flux of methionine through the methionine cycle in response to TCR activation and differentiation.

## Discussion

The present study explores how T cells control a fundamental metabolic pathway and shows that antigen receptor triggering of T cells initiates a cycle of methionine metabolism that generates the methyl donors required for RNA and histone methylation. We establish that a critical, rate limiting step to fuel protein synthesis and the methionine cycle in T cells is antigen receptor and cytokine regulation of methionine transport across the cell membrane. Naïve T lymphocytes express high levels of all the key methionine cycle enzymes but cannot transport sufficient methionine to provide the substrates for these enzymes or to fuel protein synthesis. The methionine cycle and protein synthesis are thus only initiated in T cells when antigen receptor engagement signals the expression of methionine transporters that support high rates of methionine transport. This provides some new understanding regarding mechanisms that ensure the immunological specificity of effector T cell differentiation. Only T cells that respond to antigen to upregulate methionine transport will be able to fuel protein synthesis and supply the methyl donors that permit the dynamic nucleotide methylations and epigenetic reprogramming that drives T cell differentiation.

Activated T cells express multiple candidate mammalian methionine transporters, therefore we have used a combination of pharmacological and genetic strategies and pinpoint that the sodium independent System L amino acid transporter, SLC7A5 is the dominant methionine transporter in activated T cells. Previous studies have shown that SLC7A5 is important for T cell differentiation in vitro and in vivo (*Sinclair et al., 2013*) and proposed that this reflected the role of SLC7A5 as the key leucine transporter in activated T cells and its subsequent role in regulating the activity of the leucine sensing kinase mammalian target of rapamycin complex 1 (mTORC1). The present data provide additional novel information as to why SLC7A5 is critically important for T cells; it is required to transport methionine. The present data further highlight that the importance of SLC7A5 for T cell biology reflects that it is a common transporter for multiple essential amino acids.

The dependence of T cell activation, differentiation and proliferation on extracellular methionine was striking and indicates that T cells cannot rely on methionine salvage pathways or autophagy of intracellular cargo to meet their demands for methionine. One reason methionine supply is so important for immune T cells is to fuel de novo protein synthesis. For example, we show that the expression of c-myc, a critical transcription factor for T cells, is controlled by external methionine availability and this is a simple reflection of the fact that proteins with a short half life like c-myc will be dependent on sustained availability of amino acids. In this latter context, it has been shown that in vivo activated CD8[+] T cells exhibit dynamic control of translation as they differentiate from naïve (low translation rates) into effector (high translation rates) and memory (low translation rates) cells (*Araki et al., 2017*). The present data highlight that the most critical rate limiting step for mRNA translation is the rate of methionine transport via the System L transporter SLC7A5 because it is this transporter that determines intracellular methionine bioavailability. We have previously shown that SLC7A5 expression is restricted to TCR activated effector cells and low in naïve and memory T cells (*Sinclair et al., 2013*; *Preston et al., 2015*). The dynamic changes in mRNA translation observed by *Araki et al. (2017)* could thus reflect changes in the methionine transport capacity of naïve versus effector versus memory T cells.

One other insight from present data was how much T cells depend on an external methionine supply and the controlled expression of methionine transporters to sustain the production of S-adenosylmethionine (SAM), the methyl donor for DNA, RNA and protein methyltransferases. Methionine availability and SAM production also modulate mTORC1 activity in T cells analogous to the recently described methionine/SAM sensing pathways that regulate mTORC1 in HEK 293 T cells (*Gu et al., 2017*). The supply of methyl donors for protein and nucleotide methylations is critical for many cellular processes. For example, mRNA cap methylation controls mRNA binding and translation (*Gonatopoulos-Pournatzis and Cowling, 2014*); the methylation of adenosine, 'm6A', in mRNA controls mRNA stability and is important for T cell homeostasis (*Li et al., 2017*). The data herein give new insights namely that engagement of the TCR induces RNA methylation and it is the ability of TCR triggering to control methionine transport to fuel the methionine cycle that allows the TCR to control RNA methylation. In this context the TCR does not control expression of the RNA methyltransferases; these are already present in naive T cells. The generation of methyl donors by the methionine cycle is also necessary for DNA methylations which are known to be critical for the transcriptional reprogramming that controls T cell differentiation with both increased and repressed

transcription of multiple genes, coordinated by changes in DNA methylations (*Crompton et al., 2016*; *Youngblood et al., 2017*; *Thomas et al., 2012*; *Hashimoto et al., 2013*; *Makar and Wilson, 2004*; *Ladle et al., 2016*). Regulated changes in histone methylation are also critical for T cell (*Henning et al., 2018*). Our quantitative proteomics analysis revealed that expression of histone methylation complexes is low in naïve T cells and increased upon T cell activation. The ability of immune activated T cells to control histone methylations requires the co-ordination of the expression of methionine transporters to supply the methionine cycle to make methyl donors as well as to supply methioinine for 'building' the expression of the key methyltransferase complexes. This is in contrast to the control of RNA methylation where methyltransferases responsible for RNA modifications are present in naïve T cells and poised awaiting substrate availability. The present work highlights that a full understanding of epigenetic control of cell phenotypes and the importance of RNA methylations requires knowledge of the rate limiting processes that control the methionine cycle under physiological conditions. Hence identification of methionine transporters and understanding the molecular details of how methionine transporter expression is regulated is important to understand how cells control the supply of methyl donors to support key biological processes. In this respect the importance of methionine bio-availability to other cell lineages is now recognized eg for liver cells and embryonic stem cells (*Tang et al., 2017*; *Shiraki et al., 2014*; *Mentch et al., 2015*). However, these studies do not explore or discuss the possibility that methionine availability to cells is determined by regulated expression of methionine transporters. There is only a limited understanding of the identity of the relevant methionine transporters in different cell types and very little understanding of the signals that control expression of methionine transporters in different tissues.

Finally, the importance of regulated changes in methionine transport for T cell activation predict that changes in dietary methionine, as seen by *Mentch et al. (2015)*, could impact on T cell function. Extracellular methionine availability in plasma is thus dependent upon dietary intake and dietary methionine restriction has been shown to alter histone methylation in the liver (*Mentch and Locasale, 2016*; *Mentch et al., 2015*). Whether a T cell would encounter limiting methionine concentrations is unknown, but it is a possibility. In particular it is possible that T cells may have limited access to methionine if they are positioned in tissues where other cells compete for nutrients. For example in a tumour microenvironment. The present data indicate that any study of how dietary methionine impacts any cell will need to consider the impact of restricting methionine on a range of intracellular signaling and metabolic pathways, RNA modifications and mRNA stability and not only consider the availability of methionine for protein synthesis.

# Materials and methods

## Key resources table

| Reagent type (species) or resource | Designation | Source or reference | Identifiers | Additional information |
|---|---|---|---|---|
| Genetic reagent (*M. musculus*) | JAX C57BL/6J, WT | JAX C57BL/6J Mice, Charles River UK | strain code 632 | |
| Genetic reagent (*M. musculus*) | Cd4Cre Slc7a5$^{fl/fl}$ | *Sinclair et al., 2013* | | |
| Genetic reagent (*M. musculus*) | GFP-Myc$^{KI}$ | *Huang et al., 2008*; *Preston et al., 2015* | | In these mice, a fusion protein of Myc and enhanced green fluorescent protein (GFP-Myc) is expressed from the endogenous Myc locus (GFP-Myc$^{KI}$) |
| Genetic reagent (*M. musculus*) | OT2 | *Barnden et al., 1998* | | maintained in house on a CD45.1 (LY5.1) background |
| Antibody | Anti-CD3 (armenian hamster, monoclonal) | ThermoFisher | Cat # 14-0031-82, RRID: AB_467049 | T cell activation: 1 µg/ml; Th1 differentiation: 2 µg/ml |

*Continued on next page*

*Continued*

| Reagent type (species) or resource | Designation | Source or reference | Identifiers | Additional information |
|---|---|---|---|---|
| Antibody | Anti-CD28 (syrian hamster, monoclonal) | ThermoFisher | Cat # 16-0281-82, RRID: AB_468921 | T cell activation: 2 µg/ml; Th1 differentiation: 3 µg/ml |
| Antibody | Anti-CD4 (rat, monoclonal) | Biolegend | Cat # 100530, RRID: AB_389325 | cell surface staining 1:200 |
| Antibody | Anti-TCRb (armenian hamster, monoclonal) | Biolegend | Cat # 109220, RRID: AB_893624 | cell surface staining 1:200 |
| Antibody | Anti-CD62L (rat, monoclonal) | Biolegend | Cat # 104412, RRID: AB_313099 | cell surface staining 1:200 |
| Antibody | Anti-TCR V alpha 2 (rat, monoclonal) | Biolegend | Cat # 127808, RRID: AB_1134183 | cell surface staining 1:200 |
| Antibody | Anti-CD44 (rat, monoclonal) | Biolegend | Cat # 103030, RRID: AB_830787 | cell surface staining 1:200 |
| Antibody | Anti-CD45.1 (mouse, monoclonal) | Biolegend | Cat # 110714, RRID: AB_313503 | cell surface staining 1:200 |
| Antibody | Anti-CD45.2 (mouse, monoclonal) | Biolegend | Cat # 109816, RRID: AB_492868 | cell surface staining 1:200 |
| Antibody | Anti-CD69 (armenian hamster, monoclonal) | Biolegend | Cat # 104522, RRID: AB_2260065 | cell surface staining 1:200 |
| Antibody | Anti-INFg (rat, monoclonal) | Biolegend | Cat # 505810, RRID: AB_315404 | intracellular cytokine staining 1:50 |
| Antibody | Anti-phospho-S6 (rabbit, monoclonal) | Cell Signaling Technologies | Cat # 2211, RRID: AB_331679 | intracellular staining 1:50 |
| Antibody | Anti-Histone H3 (rabbit, monoclonal) | Cell Signaling Technologies | Cat # 82241S | intracellular staining 1:50 |
| Antibody | Anti-Histone H3K4me3 (rabbit, monoclonal) | Cell Signaling Technologies | Cat # 12064 | intracellular staining 1:50 |
| Antibody | Anti-H3K27me3 (rabbit, monoclonal) | Cell Signaling Technologies | Cat # 12158 | intracellular staining 1:50 |
| Antibody | Anti-rabbit A647 | Cell Signaling Technologies | Cat # 4414, RRID: AB_10693544 | intracellular staining 1:500 |
| Antibody | Anti-mouse CD16/CD32 Fc Block, (rat, monoclonal) | BD Biosciences | Cat # 553141, RRID: AB_394656 | Fc block 1:70 |
| Peptide, recombinant protein | IL2 | Novartis, UK | Proleukin | Th1 differentiation: 20 ng/ml |
| Peptide, recombinant protein | IL12 | RnD Systems, UK | Cat # 419 ML | Th1 differentiation: 10 ng/ml |
| Peptide, recombinant protein | NP-OVA | BioSearch technolgies, UK | Cat # N-5051 | Immunisation 100 µg per mouse |

*Continued on next page*

Continued

| Reagent type (species) or resource | Designation | Source or reference | Identifiers | Additional information |
|---|---|---|---|---|
| Commercial assay or kit | EasySep CD8 positive isolation kit | STEMCELL Technologies, UK | Cat # 19853 | |
| Commercial assay or kit | SAH Elisa | Axis-Shield | Cat # FHCY100 | |
| Commercial assay or kit | m6A methylation | Epigentek | Cat # P-9005 | |
| Commercial assay or kit | m5C methylation | Epigentek | Cat # P-9009 | |
| Commercial assay or kit | Rneasy minikit | Qiagen | Cat # 74104 | |
| Chemical compound, drug | Imject Alum | Pierce, UK | Cat # 77161 | Immunisation adjuvant |
| Chemical compound, drug | OPP | Jena Bioscience | Cat # NU-931 | 20 µM, 10 min incubation |
| Chemical compound, drug | Click EdU | Carbosynth | Cat # NE08701 | 10 µM, 45 min incubation |
| Chemical compound, drug | Click EU | ThermoFisher | Cat # E10345 | 500 µM, 30 min incubation |
| Chemical compound, drug | Alexa 647 azide | ThermoFisher | Cat # A10277 | 5 µM, in Click reaction buffer |
| Chemical compound, drug | Actinomycin D | Sigma | Cat # A1410 | 5 µg/ml, 45 min |
| Chemical compound, drug | Cyclohexamide | Sigma | Cat # C7698 | 100 µg/ml, 30 min |
| Chemical compound, drug | Rapamycin | ThermoFisher | Cat # PHZ1235 | 20 nM |
| Chemical compound, drug | [3H] L-methionine | PerkinElmer | Cat # NET061 × 001MC | 1 µci/ml uptake buffer |
| Chemical compound, drug | [3H] L-phenylalanine | PerkinElmer | Cat # NET1122001MC | 0.5 µci/ml uptake buffer |
| Chemical compound, drug | [14C] L-glutamine | PerkinElmer | Cat # NEC451050UC | 0.1 µci/ml uptake buffer |
| Chemical compound, drug | silicone oil | Sigma | Cat # 175633 | layering buffer for uptake assay |
| Chemical compound, drug | dibutyl pthalate | Sigma | Cat # 524980 | layering buffer for uptake assay |
| Chemical compound, drug | Optiphase HiSafe 3 | PerkinElmer | Cat # 1200.437 | scintillant |
| Software, algorithm | FlowJo software | Treestar | | versions 9 and 10 |

## Mice and cells

C57BL/6 (wild-type, WT), *Cd4-Cre::Slc7a5*[fl/fl], GFP-Myc[KI] and OT2 TCR transgenic mice were bred and maintained in the WTB/RUTG, University of Dundee in compliance with UK Home Office Animals (Scientific Procedures) Act 1986 guidelines.

To activate primary T cells, lymph nodes were removed and disaggregated. Cells were cultured in RPMI 1640 containing L-glutamine (Invitrogen, ThermoFisher Scientific, UK), 10% FBS (Gibco, ThermoFisher Scientific, UK), 50 µM β-mercaptoethanol (β-ME, Sigma-Aldrich, UK) and penicillin/streptomycin (Gibco). Cells were stimulated with 1 µg/ml of the CD3 monoclonal antibody (2C11) and 2 µg/ml anti-CD28 (37.51; ebiosciences, ThermoFisher Scientific, UK) in the presence of cytokines IL12 (10 ng/ml; RnD Systems, UK) and 20 ng/ml IL2 (20 ng/ml; Proleukin, Novartis, UK). To generate Th1s, murine CD8[+] T cells were depleted from lymph node preparations using CD8 depletion kit (EasySep, STEMCELL Technologies, UK). The resulting mix of CD4[+] T cells and APC were cultured at $3 \times 10^5$ cells/ml for 5 days in the presence of anti-CD3 (2 µg/ml) and anti-CD28 (3 µg/ml) and cytokines IL12 (10 ng/ml) and 20 ng/ml IL2 (20 ng/ml). For proteomics samples, naïve CD4[+] T cells were isolated from lymph node and sorted gated on CD4[+], CD62L[high] and CD44[low]. Live TCR activated CD4[+] cells and Th1 cells were sorted for CD4[+] expression and DAP1 exclusion. Cells were then cultured in 100 µM or 1 µM methionine for a further 5 hr before processing for single-shot proteomics analysis.

Where indicated, methionine free RPMI (ThermoFisher) was supplemented with 10% dialysed FBS (ThermoFisher) and reconstituted with L-methionine (Sigma).

Cells were incubated at 37 ˚C with 5% $CO_2$ throughout.

## Adoptive transfers and ova immunisation

For in vivo activation and proliferation, OT2 (CD45.1) lymph node cells were injected into C57/Bl6 (CD45.2) hosts. After 24 hr, mice were immunised i.p. with 4-Hydroxy-3-nitrophenylacetyl hapten conjugated to ovalbumin (NP-OVA; 100 µg; BioSearch technologies, UK) adsorbed to alum (Pierce, UK). Spleens were harvested and transferred cells were identified and analysed at D3 after activation.

## Flow cytometry

For cell surface staining, antibodies conjugated to FITC, PE, APC, AlexaFluor 647, APC-efluor780, AlexaFluor 700, PerCPCy5.5, Brilliant Violet 421 and 605 were obtained from either BD Biosciences, eBioscience or Biolegend. Fc receptors were blocked using Fc Block (BD Biosciences). Antibody clones used were: CD4 (RM4-5), TCRβ (H57-597), Vα2 (B20.1), CD62L (MEL-14), CD44 (IM7), CD45.1 (A20), CD45.2 (104), CD69 (H1.2F3). Cells were fixed using 1% paraformaldehyde. Standard intracellular cytokine staining protocols were followed for INFγ (clone XMG1.2; Biolegend) staining.

For intracellular histone methylation staining or phospho-S6 staining, cells were permeabilised post fixation by incubation with 90% (v/v) methanol at −20°C for at least 30 min. Following permeabilisation, cells were washed twice and incubated with antibody against Tri-methyl-Histone H3 (Lys 27; clone C36B11), Tri-methyl-Histone H3 (Lys 4; clone C42D8), Histone H3 (clone D1H2 XP) or phospho-S6 Ser235/236 (# 2211) with anti-rabbit Alexa 647 (# 4414) secondary (Cell Signaling Technology). Cells were then washed and resupended in 0.5% FBS (v/v) in PBS for acquisition.

For flow cytometry assays for protein, RNA and DNA synthesis; cells were treated with either 20 µM O-propargyl-puromycin (OPP, Jena Bioscience) for 10 min, Click-iT EdU (10 µM; Thermo Fisher) for 45 mins or Click-iT EU (2 mM; Thermo Fisher) for 30 mins. The incorporation of the analogues into newly synthesized protein, RNA and DNA was measured by labelling with Alexa 647-azide (Invitrogen) using a standard Click-IT chemistry reaction (Thermo Fisher). As negative controls, cyclohexamide (100 µg/ml; Sigma; 30mins) and Actinomycin D (5 µg/ml; Sigma; 45 min) were added to stop protein and RNA synthesis, respectively.

Data were acquired on a LSR Fortessa II with DIVA software or a FACSVerse flow cytometer with FACSuite software (BD Biosciences) and analyzed using FlowJo software (TreeStar, version 9 and 10). Gating strategies are shown in Supplemental data.

## SAH measurements

S-adenosyl-homocysteine (SAH) levels were measured using a competitive ELISA (Axis Homocysteine EIA, FHCY 100, Axis-Shield) as recommended, omitting the primary enzymatic conversion of homo-cysteine to SAH step.

## Radiolabelled nutrient uptake

Briefly, nutrient uptake was carried out using $1 \times 10^6$ cells resuspended in 0.4 ml uptake medium. Each uptake for a biological replicate is performed in triplicate. Methionine uptake was carried out in HBSS (ThermoFisher Scientific) containing [$^3$H] L-methionine (1 µci/ml) and a final extracellular L-methionine concentration of 0.5 µm. Similarly, phenylalanine uptake was performed using [$^3$H] L-phenylalanine (0.5 µci/ml) and a final extracellular L-leucine concentration of 0.5 µm. 2 min uptake assays were carried out layered over 0.5 ml of 1:1 silicone oil (Dow Corning 550 (BDH silicone products; specific density, 1.07 g/ml):dibutyl phthalate (Fluka). Where indicated, [$^{14}$C] L-glutamine (0.1 µci/ml) was added to assay glutamine uptake simultaneously. Cells were pelleted below the oil, the aqueous supernatant solution, followed by the silicon oil/dibutyl phthalate mixture was aspirated, and the cell pellet underneath resuspended in 200 µl NaOH (1M) and β-radioactivity measured by liquid scintillation counting in a Beckman LS 6500 Multi-Purpose Scintillation Counter (Beckman Coulter). Where indicated, 5 mM BCH, L-Alanine, L-Lysine, L-Leucine, L-Methionine or MeAIB were used respectively to quench radiolabeled ligand uptake. The sodium free buffer was TMACl as described in *Baird et al. (2009)*. Data is expressed as molecules radiotracer per cell per minute. [$^3$H] L-methionine, [$^3$H] L-phenylalanine and [$^{14}$C] L-glutamine were obtained from Perkin Elmer. All other chemicals were obtained from Sigma.

## $^3$H- methionine incorporation

IL2/12 maintained effector Th1 cells were cultured with $^3$H-methionine for 6 hr. Protein from $5 \times 10^6$ cells was precipitated with 0.5 ml 10% trichloroacetic acid (TCA), for 15mins at room temperature. The protein pellet was washed (x3) with cold acetone, and acetone was allowed to evaporate. RNA was isolated from $5 \times 10^6$ cells with RNeasy minikit (Qiagen), and quantified by NanoDrop (Thermo-Fisher). Samples were resuspended in scintillation fluid (Optiphase HiSafe 3, PerkinElmer) and $^3$H radioactivity in TCA precipitated protein, or RNA was measured by liquid scintillation counting in a Beckman LS 6500 Multi-Purpose Scintillation Counter (Beckman Coulter).

m6A and m5C RNA methylation m6A and m5C RNA methylation was quantified using the fluoro-metric EpiQuik m6A or 5-mC RNA Methylation quantification Kit (Epigentek) respectively, following RNA isolation from $5 \times 10^6$ cells using RNAEasy minikit (Qiagen).

## Metabolomics:

### Metabolite extraction-

Metabolite extraction was performed as described in previous study (*Liu et al., 2015*). The supernatant was transferred to a new Eppendorf tube and dried in vacuum concentrator at room temperature. The dry pellets were reconstituted into 30 µl sample solvent (water:methanol:acetonitrile, 2:1:1, v/v) and 3 µl was further analyzed by liquid chromatography-mass spectrometry (LC-MS).

### LC-MS method-

Ultimate 3000 UHPLC (Dionex) is coupled to Q Exactive-Mass spectrometer (QE-MS, Thermo Scientific) for metabolite profiling. A hydrophilic interaction chromatography method (HILIC) employing an Xbridge amide column (100 × 2.1 mm i.d., 3.5 µm; Waters) is used for polar metabolite separation. Detailed LC method was described previously (*Liu et al., 2014b*) except that mobile phase A was replaced with water containing 5 mM ammonium acetate (pH 6.8). The QE-MS is equipped with a HESI probe with related parameters set as below: heater temperature, 120°C; sheath gas, 30; auxiliary gas, 10; sweep gas, 3; spray voltage, 3.0 kV for the positive mode and 2.5 kV for the negative mode; capillary temperature, 320°C; S-lens, 55; scan range (*m/z*): 70 to 900; resolution: 70000; automated gain control (AGC), $3 \times 10^6$ ions. Customized mass calibration was performed before data acquisition.

## Metabolomics data analysis

LC-MS peak extraction and integration were performed using commercial available software Sieve 2.2 (Thermo Scientific). The peak area was used to represent the relative abundance of each metabolite in different samples. The missing values were handled as described in previous study (*Liu et al., 2014b*).

## Proteomics

## Naïve CD4$^+$ and effector Th1 (TMT labelled)

### Sample preparation and TMT labelling

Cell pellets were lysed in 400 µL lysis buffer (4% SDS, 50 mM TEAB pH 8.5, 10 mM TCEP). Lysates were boiled and sonicated with a BioRuptor (30 cycles: 30 s on, 30 s off) before alkylation with iodoacetamide for 1 hr at room temperature in the dark. The lysates were subjected to the SP3 procedure for protein clean-up (*Hughes et al., 2014*) before elution into digest buffer (0.1% SDS, 50 mM TEAB pH 8.5, 1 mM CaCl$_2$) and digested with LysC and Trypsin, each at a 1:50 (enzyme:protein) ratio. TMT labelling and peptide clean-up were performed according to the SP3 protocol. Samples were eluted into 2% DMSO in water, combined and dried in vacuo.

### Basic reverse-phase fractionation

The TMT samples were fractionated using off-line high pH reverse phase chromatography: samples were loaded onto a 4.6 × 250 mm Xbridge BEH130 C18 column with 3.5 µm particles (Waters). Using a Dionex BioRS system, the samples were separated using a 25 min multistep gradient of solvents A (10 mM formate at pH 9 in 2% acetonitrile) and B (10 mM ammonium formate pH 9 in 80% acetonitrile), at a flow rate of 1 mL/min. Peptides were separated into 48 fractions which were consolidated into 24 fractions. The fractions were subsequently dried and the peptides redissolved in 5% formic acid and analysed by LC-MS.

### Liquid chromatography electrospray tandem mass spectrometry analysis (LC-ES-MS/MS)

1 µg per fraction was analysed using an Orbitrap Fusion Tribrid mass spectrometer (Thermo Scientific) equipped with a Dionex ultra high-pressure liquid chromatography system (nano RSLC). RP-LC was performed using a Dionex RSLC nano HPLC (Thermo Scientific). Peptides were injected onto a 75 µm × 2 cm PepMap-C18 pre-column and resolved on a 75 µm × 50 cm RP- C18 EASY-Spray temperature controlled integrated column-emitter (ThermoFisher) using a four hour multistep gradient from 5% B to 35% B with a constant flow of 200 nL min$^{-1}$. The mobile phases were: 2% ACN incorporating 0.1% FA (Solvent A) and 80% ACN incorporating 0.1% FA (Solvent B). The spray was initiated by applying 2.5 kV to the EASY-Spray emitter and the data were acquired under the control of Xcalibur software in a data dependent mode using top speed and 4 s duration per cycle, the survey scan is acquired in the Orbitrap covering the *m/z* range from 400 to 1400 Th with a mass resolution of 120,000 and an automatic gain control (AGC) target of 2.0 e5 ions. The most intense ions were selected for fragmentation using CID in the ion trap with 30% CID collision energy and an isolation window of 1.6 Th. The AGC target was set to 1.0 e4 with a maximum injection time of 70 ms and a dynamic exclusion of 80 s. During the MS3 analysis for more accurate TMT quantifications, 10 fragment ions were co-isolated using synchronous precursor selection using a window of 2 Th and further fragmented using HCD collision energy of 55%. The fragments were then analysed in the Orbitrap with a resolution of 60,000. The AGC target was set to 1.0 e5 and the maximum injection time was set to 300 ms.

### Database searching and reporter ion quantification

The data were processed, searched and quantified with the MaxQuant software package, version 1.5.8.3, Proteins and peptides were identified using the UniProt *mouse* reference proteome database (SwissProt and Trembl) and the contaminants database integrated in MaxQuant using the Andromeda search engine (*Cox and Mann, 2008*; *Cox et al., 2011*) with the following search parameters: carbamidomethylation of cysteine and TMT modification on peptide N-termini and lysine side chains were fixed modifications, while methionine oxidation, acetylation of N-termini of proteins. The false discovery rate was set to 1% for positive identification of proteins and peptides

with the help of the reversed mouse Uniprot database in a decoy approach. Copy numbers were calculated as described (*Wiśniewski et al., 2014*) after allocating the summed MS1 intensities to the different experimental conditions according to their fractional MS3 reporter intensities.

## TCR activated CD4$^+$ (label free)

Cell pellets were lysed and peptides generated using the SP3 method as described above, but without TMT labelling. After elution with DMSO samples were fractionated by high pH reverse phase chromatography as described above but with the following modifications; samples were loaded onto a 2.1 × 150 mm Xbridge BEH130 C18 column with 3.5 μm particles (Waters) on a UltiMate 3000 HPLC (Thermo Fisher Scientific) and separated at a flow rate of 0.3 mL/min. Peptides were separated into 16 fractions which were consolidated into eight fractions. The fractions were subsequently dried and the peptides redissolved in 5% formic acid. LC-MS analysis was performed as described previously (*Rollings et al., 2018*) with slight modifications. Samples were injected onto a nanoscale C18 reverse-phase chromatography system (UltiMate 3000 RSLC nano, Thermo Fisher Scientific) before being electrosprayed into a Orbitrap mass spectrometer (LTQ Orbitrap Velos Pro; Thermo Fisher Scientific). The chromatography buffers used were as follows: HPLC buffer A (0.1% formic acid), HPLC buffer B (80% acetonitrile and 0.08% formic acid), and HPLC buffer C (0.1% formic acid). Peptides were loaded onto an Acclaim PepMap100 nanoViper C18 trap column (100 μm inner diameter, 2 cm; Thermo Fisher Scientific) in HPLC buffer C with a constant flow of 5 μl/min. After trap enrichment, peptides were eluted onto an EASY-Spray PepMap RSLC nanoViper, C18, 2 μm, 100 Å column (75 μm, 50 cm; Thermo Fisher Scientific) using the following buffer gradient: 2% B (0 to 6 min), 2% to 35% B (6 to 130 min), 35% to 98% B (130 to 132 min), 98% B (132 to 152 min), 98% to 2% B (152 to 153 min), and equilibrated in 2% B (153 to 170 min) at a flow rate of 0.3 μl/min. The eluting peptide solution was automatically electrosprayed into the Orbitrap mass spectrometer (LTQ Orbitrap Velos Pro; Thermo Fisher Scientific) using an EASY-Spray nanoelectrospray ion source at 50°C and a source voltage of 1.9 kV (Thermo Fisher Scientific). The mass spectrometer was operated in positive ion mode. Full-scan MS survey spectra (mass/charge ratio, 335 to 1800) in profile mode were acquired in the Orbitrap with a resolution of 60,000. Data were collected using data-dependent acquisition: the 15 most intense peptide ions from the preview scan in the Orbitrap were fragmented by collision-induced dissociation (normalized collision energy, 35%; activation Q, 0.250; activation time, 10 ms) in the LTQ after the accumulation of 5000 ions. Precursor ion charge state screening was enabled, and all unassigned charge states as well as singly charged species were rejected. The lock mass option was enabled for survey scans to improve mass accuracy. MS data was analysed as described above with the following modifications; MaxQuant software package, version 1.6.0.1. Proteins and peptides were identified using a uniport mouse canonical plus isoforms database (2$^{nd}$ August 2018).

## Single-shot proteomics on methionine deprived Th1 cells

Cell pellets were processed as described above, label free. After elution of peptides with DMSO, samples were processed by single shot LC-MS. Analysis of peptides was performed on a Q-exactive-HFX (Thermo Scientific) mass spectrometer coupled with a Dionex Ultimate 3000 RS (Thermo Scientific). LC buffers were the following: buffer A (0.1% formic acid in Milli-Q water (v/v)) and buffer B (80% acetonitrile and 0.08% formic acid in Milli-Q water (v/v). Aliquots of 15 μL of each sample were loaded at 10 μL/min onto a trap column (100 μm × 2 cm, PepMap nanoViper C18 column, 5 μm, 100 Å, Thermo Scientific) equilibrated in 2% buffer B. The trap column was washed for 5 min at the same flow rate and then the trap column was switched in-line with a Thermo Scientific, resolving C18 column (75 μm × 50 cm, PepMap RSLC C18 column, 2 μm, 100 Å). The peptides were eluted from the column at a constant flow rate of 300 nl/min with a linear gradient from 5% buffer B to 35% buffer B in 120 min, and then to 98% buffer B by 122 min. The column was then washed with 98% buffer B for 15 min and re-equilibrated in 2% buffer B for 17 min. Q-exactive HFX was used in data dependent mode. A scan cycle comprised MS1 scan (m/z range from 335 to 1800, with a maximum ion injection time of 50 ms, a resolution of 60,000 and automatic gain control (AGC) value of 3 × 106) followed by 40 sequential dependant MS2 scans (with an isolation window set to 1.4 Da, resolution at 7500, maximum ion injection time at 50 ms and AGC 1 × 105. MS data was analysed as described for the label-free CD4$^+$ TCR activated sample above. Estimates of protein copy number

were used to calculate the fold change of protein abundance between 100 µM and 1 µM methionine and a two-sample T test with unequal variance was performed to identify proteins significantly changing.

## Public Availability of Data

The mass spectrometry proteomics data have been deposited to the ProteomeXchange Consortium data repository (https://www.ebi.ac.uk/pride/archive/login) and can be accessed with identifier PXD012052 (for TCR activated CD4 proteome), PXD012053 (for methionine restricted Th1 proteome) and PXD012058 (for naïve and effector CD4 (Th1) proteomes).

## Statistics

Statistical analyses were performed using Prism 4.00, GraphPad Software, or Sigma Plot (Systat). A Shapiro-Wilk test for normality was performed to determine suitable tests for parametric or non-parametric populations. F-tests were performed to determine equal variance of populations, otherwise tests assuming unequal variance were performed. Multiple comparisons in one-way ANOVA analyses were corrected for using the Holm-Sidak method. Where indicated, EC50 and $R^2$ values were calculated with least squares fit, no constraints applied. All used tests are stated in the respective figure legends, statistical tests used for proteomics analysis are stated in the Proteomics Materials and methods section.

# Acknowledgements

We thank Victoria Cowling (University of Dundee, UK) and the Cantrell group members for their critical discussion of the data, the Biological Resources unit and the Flow Cytometry facility (A Whigham and R Clarke) at the University of Dundee. This work was supported by the Wellcome Trust (Principal Research Fellowship to DAC 097418/Z/11/Z and 205023/Z/16/Z; Wellcome Trust Equipment Award 202950/Z/16/Z).

# Additional information

## Funding

| Funder | Grant reference number | Author |
|---|---|---|
| Wellcome Trust | 105024/Z/14/Z (Wellcome Trust Strategic Award) | Angus I Lamond Doreen A Cantrell |
| Wellcome Trust | 202950/Z/16/Z (Wellcome Trust Equipment Award) | Doreen A Cantrell |
| Wellcome Trust | 205023/Z/16/Z (Principal Research Fellowship) | Doreen A Cantrell |

The funders had no role in study design, data collection and interpretation, or the decision to submit the work for publication.

## Author contributions

Linda V Sinclair, Conceptualization, Formal analysis, Supervision, Validation, Investigation, Visualization, Methodology, Writing—original draft, Project administration, Writing—review and editing; Andrew JM Howden, Formal analysis, Investigation, Visualization, Writing—original draft, Writing—review and editing; Alejandro Brenes, Formal analysis, Visualization, Writing—review and editing; Laura Spinelli, Investigation, Visualization, Writing—review and editing; Jens L Hukelmann, Formal analysis, Investigation, Writing—review and editing; Andrew N Macintyre, Xiaojing Liu, Investigation, Writing—review and editing; Sarah Thomson, Investigation; Peter M Taylor, Writing—review and editing; Jeffrey C Rathmell, Resources, Writing—review and editing; Jason W Locasale, Resources, Methodology, Writing—review and editing; Angus I Lamond, Resources, Methodology, Writing—review and editing, Funding acquisition; Doreen A Cantrell, Conceptualization, Resources, Funding acquisition, Writing—original draft

## Author ORCIDs
Linda V Sinclair (ID) http://orcid.org/0000-0003-1248-7189
Andrew JM Howden (ID) http://orcid.org/0000-0002-4332-9469
Alejandro Brenes (ID) http://orcid.org/0000-0001-8298-2463
Jason W Locasale (ID) https://orcid.org/0000-0002-7766-3502
Angus I Lamond (ID) https://orcid.org/0000-0001-6204-6045
Doreen A Cantrell (ID) https://orcid.org/0000-0001-7525-3350

## Ethics

Animal experimentation: All animal experiments were performed under Project License PPL 60/4488 and P4BD0CE74.The University of Dundee Welfare and Ethical Use of Animals Committee accepted the project licence for submission to the HO. All studies, breeding and maintenance performed in Dundee in compliance with UK Home Office Animals (Scientific Procedures) Act 1986 guidelines. Individual study plans were approved and deemed compliant by the UVS/Named Compliance Officer.

## Decision letter and Author response

Decision letter https://doi.org/10.7554/eLife.44210.020
Author response https://doi.org/10.7554/eLife.44210.021

# Additional files

## Supplementary files

• Supplementary file 1. Flow cytometry plots showing representative gating strategies for flow data shown in *Figures 1* and *2*.
DOI: https://doi.org/10.7554/eLife.44210.011

• Transparent reporting form
DOI: https://doi.org/10.7554/eLife.44210.012

## Data availability

All data generated or analysed during this study are included in the manuscript and supporting files, or have been submitted to the PRIDE ProteomeXchange consortium under Project IDs PXD012052, PXD012053 and PXD012058.

The following datasets were generated:

| Author(s) | Year | Dataset title | Dataset URL | Database and Identifier |
|---|---|---|---|---|
| Linda V Sinclair, Andrew JM Howden, Alejandro Brenes, Laura Spinelli, Jens L Hukelmann, Andrew N Macintyre, Xiaojing Liu, Sarah Thomson, Peter M Taylor, Jeffrey C Rathmell, Jason W Locasale, Angus I Lamond, Doreen A Cantrell | 2019 | Methionine restricted Th1 proteome | https://www.ebi.ac.uk/pride/archive/projects/PXD012053 | PRIDE, PXD012053 |
| Linda V Sinclair, Andrew JM Howden, Alejandro Brenes, Laura Spinelli, Jens L Hukelmann, Andrew N Macintyre, Xiaojing Liu, Sarah Thomson, Peter M Taylor, Jeffrey C Rathmell, | 2019 | Naïve and effector CD4 (Th1) proteomes | https://www.ebi.ac.uk/pride/archive/projects/PXD012058 | PRIDE, PXD012058 |

Jason W Locasale,
Angus I Lamond,
Doreen A Cantrell

| | | | | |
|---|---|---|---|---|
| Linda V Sinclair, Andrew JM Howden, Alejandro Brenes | 2019 | TCR activated CD4 proteome | https://www.ebi.ac.uk/pride/archive/projects/PXD012052 | PRIDE, PXD012052 |

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
