## [Decision Letter]

Thank you for submitting your article "Antigen receptor control of methionine metabolism in T cells" for consideration by *eLife*. Your article has been reviewed by three peer reviewers, and the evaluation has been overseen by a Reviewing Editor and Tadatsugu Taniguchi as the Senior Editor. The following individual involved in review of your submission has agreed to reveal their identity: Bing Su.

The reviewers have discussed the reviews with one another and the Reviewing Editor has drafted this decision to help you prepare a revised submission.

Summary:

Sinclair and colleagues use a variety of approaches, including mass spec and metabolic labeling to show that T cells use methionine to produce methyl donors important for protein synthesis and nucleotide methylations, and RNA and histone methylations. The work is important as it provides the reader with a clear picture of how this amino acid is utilized in T cells. Their results establish that methionine uptake is controlled at the level of transporter expression, rather than enzyme expression, which is interesting. Studies such as these add to our understanding of how metabolism in T cells directs their activation, differentiation, and function. The data are convincing and the arguments are well supported.

Essential revisions:

1) The authors should extend their discussion of why methionine used to methylate DNA, which is largely thought of as repressive, is required for T cell activation. Do they think these methylations are on repressors of T cell activation?

2) What happens to mTOR activation, and Myc, in the absence of methionine? One would assume that mTOR signals drop in the absence of methionine. These data are important to include and/or discuss as the authors' previous work showed that leucine uptake via SLC7A5 is required for mTOR activation and myc expression. It would be helpful for the reader to know whether both leucine and methionine are required, and if so, what about other amino acids? Which amino acids do they think are critical, or are they all critical? If they are all critical, do they think they actually vary in what they supply to the cell? Is mTORC1 activity required for methionine uptake? Some discussion would be helpful.

3) The authors conclude that the concentration of methionine required for protein synthesis was 1.39μM, 2.94μM for RNA, and 12.62μM for DNA in vitro. The authors perhaps should discuss what the level of methionine is in the blood. If it is around 2-10μM, rather than much higher, then it might be that in vivo, it is not only transporter expression, but availability of extracellular methionine could determine uptake. The authors should comment on the physiological conditions where a T cell would encounter limiting concentrations of methionine.

4) Can a link with methionine metabolism to other forms of T cell metabolism, such as glucose metabolism or lipid metabolism, be made to put the methionine data into context.

---

## [Author Response]

Essential revisions:1) The authors should extend their discussion of why methionine used to methylate DNA, which is largely thought of as repressive, is required for T cell activation. Do they think these methylations are on repressors of T cell activation?

We thank the reviewers for this excellent suggestion and have modified the Discussion to include some discussion about DNA methylation.

The salient point is that T cell differentiation involves a quite remarkable transcriptional reprogramming with both increased and repressed transcription of many genes. There are many studies showing that transcriptional silencing mediated by DNA methylation is essential for peripheral T cell differentiation. It is also evident that epigenetic repression programs in T cells are dynamic and reversible (Crompton et al., 2015; Youngblood et al., 2017; Thomas et al., 2012; Hashimoto et al., 2013; Makar and Wilson, 2004). T cell proliferative expansion and differentiation is moreover sensitive to loss of the core DNA methyl transferase DNMT1 (Chappell et al., 2006). The present study did not explore DNA methylation although it is clear that dynamic changes in DNA methylation mediated by DNA methyltransferases are critical for peripheral T cell proliferation and differentiation in response to antigen. Accordingly, the supply of methyl donors for these key epigenetic changes via regulation of methionine transport would be critical for the generation of an optimal immune response.

2) What happens to mTOR activation, and Myc, in the absence of methionine? One would assume that mTOR signals drop in the absence of methionine. These data are important to include and/or discuss as the authors' previous work showed that leucine uptake via SLC7A5 is required for mTOR activation and myc expression. It would be helpful for the reader to know whether both leucine and methionine are required, and if so, what about other amino acids? Which amino acids do they think are critical, or are they all critical? If they are all critical, do they think they actually vary in what they supply to the cell? Is mTORC1 activity required for methionine uptake? Some discussion would be helpful.

We have now investigated the impact of methionine deprivation on mTORC1 activity and Myc expression and now include these data in a new figure (Figure 6, subsection “The impact of methionine restriction on c-myc expression and mTORC1 activity”). The new data show that mTORC1 activity in T cells is sensitive to methionine deprivation although the impact of methionine deprivation is less than the effect of total amino acid deprivation (new Figure 6). We also show that the expression of Myc in T cells is partially sensitive to methionine deprivation (new Figure 6). We have shown previously that Myc expression and mTORC1 activity in T cells are totally dependent on the expression of SLC7A5 which transports leucine and, as shown in the present paper, methionine. The fact that Myc expression and mTORC1 activity are only partially sensitive to methionine deprivation tells us that the delivery of leucine by SLC7A5 is also important. We have modified the Discussion (second paragraph) to clarify these thoughts. In short we think that SLC7A5 is crucial because it supplies both methionine and leucine and other essential amino acids.

One final point in this context, the reviewers ask: Is mTORC1 activity required for methionine uptake?

No, we have never seen an example of any mTORC1 control of amino acid transport in T cells and we have looked at this extensively in proteomics data that are as yet unpublished.

These new data regarding Myc expression and mTORC1 activity are presented as a new figure (Figure 6) and new Results section, and incorporated into the Discussion. We feel that the addition of this data significantly furthers our understanding of the role(s) of methionine for T cells and thank the reviewers (again) for this constructive avenue.

3) The authors conclude that the concentration of methionine required for protein synthesis was 1.39μM, 2.94μM for RNA, and 12.62μM for DNA in vitro. The authors perhaps should discuss what the level of methionine is in the blood. If it is around 2-10μM, rather than much higher, then it might be that in vivo, it is not only transporter expression, but availability of extracellular methionine could determine uptake. The authors should comment on the physiological conditions where a T cell would encounter limiting concentrations of methionine.

We agree with the reviewers on the importance of knowing the levels of in vivo methionine availability i.e. true physiological conditions. Here the human Metabolome Database indicates the healthy (human) adult range in plasma to be 21.0- 42.9μM. This data is collated from several sources (Wishart et al., 2018; Psychogios et al., 2011). Mentch et al. measured and described the range of plasma methionine levels in fasted humans to be 3-30μM (Mentch et al., 2015), and subsequent analysis showed that this variation in range was significantly impacted by dietary factors. Of note here is that this range, achieved with dietary variation, *does* reflect the range at which methionine restriction has an impact on T cells (i.e. the IC50 values determined above). Conditions in which components of the methionine cycle are disrupted (e.g. methyltetrahydrofolate reductase (MTHFR) deficiency, homocystinuria, cobalamin deficiencies) show dramatically lower blood methionine concentrations (<10μM) (Wishart et al., 2018; Psychogios et al., 2011). The Human Metabolome Database also has metabolite measurements on cerebrospinal fluid (CSF). Normal methionine levels in CSF are approximately 5μM.

Whether a T cell would encounter limiting methionine concentrations has not been measured but is clearly possible and here we would say that very little is known about methionine levels in tissues or in microenvironments where different cells might compete for the same nutrient. Here it is pertinent that a recent study measured glucose levels to be 10-fold lower in solid tumour than in plasma (Ho et al., 2015), extrapolating this to other nutrients would put methionine well within a limiting range. It is also important to note that transport of methionine is always in competition with other amino acid substrates of SLC7A5. The Km for methionine transport is 20μM – approximately the same as tryptophan (21.4μM), and leucine (19.7μM). With this in mind, serum levels of these substrates vary substantially: mouse serum levels of methionine are 20μM (human 30μM), leucine is 100μM (human 150μM), tryptophan is 50μM (human 50-80μM) (mouse values from She et al., 2007, human values from Wishart et al., 2018). We have expanded our Discussion to highlight possible physiological conditions where a T cell would encounter limiting concentrations of methionine, and the consequential competition relating to transporter expression and substrate availability. However, we feel that the expression of the methionine transporter by activated T cells is a master controller as to whether the cell has the potential/capacity to fully engage the methionine pathway.

4) Can a link with methionine metabolism to other forms of T cell metabolism, such as glucose metabolism or lipid metabolism, be made to put the methionine data into context.

Whether methionine metabolism links in to other forms of T cell metabolism is a very interesting question/line of thought. Whilst glucose metabolism and lipid metabolism have not been directly explored in this study, the additional new data regarding the effect of methionine depletion on mTORC1 activity and c-myc expression would strongly predict a cascade of effects on these metabolic pathways.

– mTORC1 activity – promotes ribosome biogenesis and protein translation; regulates lipid metabolism, including de novo biogenesis via SREBPs; mTORC1 activity is required to sustain high levels of glycolysis in effector T cells (Hukelmann et al., 2016; Finlay et al., 2012). Mediates glycolytic control effects through HIF1a in effector cells (Finlay et al., 2012).

– c-myc expression – regulates glycolytic and glutaminolysis programs in activated T cells, critical for driving the metabolic “switch” seen in T cell activation (Wang et al., 2011).

We have expanded the Discussion accordingly to highlight the complex interplay with these different metabolic systems in T cells. We feel that this has significantly improved the paper, and (thankfully) acknowledge the reviewers’ line of questions which pushed this forward.